# The Precarious Survival of an Ancient Cultural Landscape: The Thousand-Year-Old Olive Trees of the Valencian Maestrat (Spain)

**Joan Carles Membrado-Tena** * and **Jorge Hermosilla-Pla** *

ESTEPA Research Group *Estudios del Territorio, Paisaje y Patrimonio* (Studies on Territory, Landscape and Heritage), Department of Geography, Universitat de València, Av. Blasco Ibañez 28, 46010 València, Spain

* Correspondence: joan.membrado@uv.es (J.C.M.-T.); jorge.hermosilla@uv.es (J.H.-P.)

**Abstract:** The object of study of this article is the Valencian Maestrat olive growing system (eastern Spain). Its landscape and heritage values are evaluated through a qualitative assessment method based on a Spanish research project studying MTASs (Multifunctional, Territorialized Agrifood Systems), which can be described as an alternative agricultural model to the worldwide agro-industrial model. The results of this analysis show that this olive growing system coincides with the MTAS criteria as regards the landscape, which offers ecosystem services (food, structured soil, and absorption of $CO_2$ emissions) and possesses cultural and heritage values (ancient olive trees, traditional rain-fed lands, unaltered plot structures, and dry stone structures). However, as far as production is concerned, the Maestrat olive growing system does not respond fully to MTAS principles: its particular environmental conditions (soil and climate) restrict the production of quality oil, which is processed and marketed mainly through cooperatives and is economically viable only thanks to CAP (Common Agricultural Policy) aid. Nevertheless, the cooperative system allows for the survival, albeit precarious, of this agricultural system. Only a small number of Maestrat olive growing farmers produce quality oil in accordance with MTAS criteria (such as local single varieties, unique flavour, proximity sales, territory closeness, or good farming practices).

**Keywords:** cultural landscape; Maestrat (Spain); olive growing system; MTAS; evaluation method

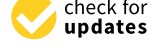



## 1. Introduction

A cultural landscape can be defined as a result of the physical environment being altered by social and economic forces over time [1]. Cultural landscapes have been afforded increasing attention with the influence of globalization and the notion of sustainable development. The progressive abandonment of farmlands with their associated landscapes has changed land structures and functions while accelerating the degradation of cultural landscapes and food production associated with them [2,3]. The purpose of this article is to analyse the Maestrat olive growing cultural landscape, which is a district to the north of the Region of València (Spain) (Figures 1–3). This analysis is implemented according to an evaluation method used to define an MTAS (Multifunctional, Territorialized Agrifood System), also known as SAMUTER (a Spanish acronym meaning *Sistema Agrario Multifuncional y Territorializado*). An MTAS is a hybrid system that combines tradition, multifunction, governance, and sustainable practices as a basis for territorial development. In contrast to hegemonic agro-industrial systems, oriented towards large-scale food production and without territorial characteristics, an MTAS claims its identity from unique productions derived from more sustainable practices based on particular environmental conditions that provide a valuable landscape with character [4,5].

An MTAS embraces some of the Sustainable Development Goals (SDGs) of the United Nations 2030 Agenda, such as proximity markets (goal 15.9), family farming (8.3, 8.4), and participatory governance where agents, companies, and institutions are attached to the

territory through productive differentiation and maintenance of a unique landscape (12.8), which results in sustainable cultural tourism that generates employment and consumes local products (8.9) [6].

Various schools of thought have addressed the issue of the relationships between territory and local food production systems, relying on the discourse of productive sustainability and preservation of a landscape that identifies and characterizes the territory [7–9]. Food production is shaped by the nature of the landscape and people's relationship with it. Analysing food as well as landscapes can reveal much about the relationships that both create, particularly about their symbolic and cultural significance [10–12]. Food can be viewed as a representation and manifestation of human identity, and this is also reflected in the character of the landscape. Food identity becomes rooted in the physical form of our landscapes [13]. There is increasing interest in the way landscapes are shaped by the production of food, in particular how local food cultures evolve from the interactions between people and the food that is grown as a result of particular environmental conditions [14].

Concerned about this relationship between food and landscape, the European Union stimulates and rewards, through the new 2023–2027 CAP (Common Agricultural Policy), landscape protection through a new payment linked to its preservation [15]. Beyond the EU and other institutions, such as the World Bank through some of its projects, which are interested in the relationship between food and landscape, a growing number of cultural landscapes recognized by UNESCO [16] show this increasing interest in landscapes shaped by food production: Honghe Hani Rice Terraces in China, Coffee Landscape in Colombia, Agave and Tequila Landscape in Mexico, and Olives and Vines in Palestine [17]. UNESCO protection of wine landscapes is especially significant: Alto Douro and Pico Island in Portugal, Champagne and Burgundy in France, Piedmont and Treviso in Italy, Tokaj in Hungary, and Wachau in Austria. Camaioni et al. (2016) enhance the concept of 'terroir' and the quality of food, identity, and landscape in an examination of vineyard landscapes [18]. Also regarding wine, Flinzberger et al. consider the territory as a distinctive basis for its quality and character, and territory becomes a landscape when the prestige achieved by the product claims it as a representative image [19]. Martínez-Arnáiz et al. say that the territory has to amplify its referential discourse, selling consumers both the *terroir* and its image: the landscape. This territorial enhancement through wine tourism reflects the efforts made through landscape preservation investments [6].

Beyond the recognition of food landscapes as UNESCO heritage, the Food and Agriculture Organization (FAO) has established a program called Globally Important Agricultural Heritage Systems (GIAHS). GIAHS landscapes are characterized by a long history, presence of traditional practices, typical foods, complex landscape mosaics, and high biocultural diversity [20]. Since 2005, the FAO has designated 74 systems in 24 countries as GIAHs, five of them in Spain. Among them is one halfway between Catalonia and València: Ancient Olive Trees of the Sénia Territory. Part of its Valencian area—known as Maestrat—is the region of study for this paper (Figure 2).

Among Mediterranean landscapes, the olive tree, together with vines and wheat, forms a trilogy that dates back to antiquity [20]. Spain currently has the largest area of olive groves in the world and produces more than half of the world's olive oil [21]. In Roman times, the province of Baetica was already known for its olive oil production around the Valley of the River Guadalquivir (called *Baetis* in Latin, and that is why the province was called *Baetica*) [22]. This area is still the world's largest oil supplier today, but the fertile Baetic countryside—today a part of Andalusia—is not the only one where oil production dates back to ancient times. On the eastern Iberian Mediterranean slopes, there are some dry and stony regions, generally less productive than those of Andalusia, where the olive growing tradition dates back at least the last two millennia. One of these regions is the above-mentioned Maestrat, in the north of València, neighbouring southern Catalonia and sharing with it the same olive growing system (Figure 1).

In the agricultural system of Maestrat—a region of rough soils and dry land—olive trees have probably been cultivated since the time of the Iberians (around 500 BCE) [23–25].

In ancient times, its crops were, where the cold did not prevent it, those of the Mediterranean trilogy and so remained in Roman, Islamic, medieval Christian, and modern times and until the second half of the twentieth century, when wheat and vines were discarded as unproductive and thus replaced by the more lucrative olive trees. Maestrat—and the adjacent Catalan region of Montsià—have the largest accumulation of thousand-year-old olive trees in the Mediterranean basin and the rest of the world (Figure 2) [25]. There, the empirical presence of the olive tree dates at least as far back as the medieval Islamic period according to the surviving thousand-year-old olive trees, although, in Roman and Iberian times, they should have grown too, well-adapted to the soil conditions of the area. This region, which has transitioned from traditional polyculture to olive monoculture, manages most of its olive production through cooperatives via agro-industrial processes and sale in medium-distance markets (not in proximity markets). Only a minority of producers seek sustainable practices, territorial quality brands, and commercial proximity and are in line with the MTAS philosophy: agricultural areas in which one or more crops are deeply rooted in the place, giving it character and generating a network of production, distribution, and consumption relations as an alternative to agro-industrial systems [6].

More and more academic and social voices are pointing out that the current global agrifood system is unsustainable because it is unfair to producers and has a high environmental impact, and they advocate fairer and healthier food systems that are linked to the territory [26–28].

The main aim of this article is to find out whether the Maestrat olive growing system corresponds to the definition of an MTAS. It must be said that, on the one hand, it preserves its traditional olive growing landscape, but, on the other, it has been expanded in recent decades to the detriment of cereals and vines; it produces mostly low-quality oil that is managed by local cooperatives and consumed in distant agro-industrial markets. However, part of its production is also high-quality (extra virgin), single-variety (local varieties), of proximity (sold mainly in the Valencian area), environmentally healthy (good agricultural practices), attached to the territory (better preservation of the landscape), and creates synergies with the social fabric (using cooperative machinery).

## 2. Methods and Case Study

This article derives from the MTAS (SAMUTER) project, funded by Spain's Ministry of Science, Innovation and Universities, and it is applied to the Valencian region of Maestrat. The working method used for the results of this article derives from the literature review, fieldwork, mapping, interviews, and a method for assessing landscape and heritage quality, developed by Mayordomo-Maya and Hermosilla-Pla [29,30].

Significant qualitative information was obtained from semi-structured interviews. The questions were agreed upon by the MTAS working group in the Department of Geography at the University of València. The ten questions asked were (1) Do you think that this territory conforms to a MTAS? (2) If yes, who are the main actors in this MTAS? (3) Why does this system differ from others? (4) Does this system preserve its original landscape? (5) What traditional techniques and knowledge distinguish this system from others? (6) Is the current system environmentally sustainable? Is it socioeconomically sustainable? (7) What elements beyond farming make this system a multifunctional space? (8) Do the governmental authorities make investments to preserve this system? (9) Is there social identification with this system? (10) What are the main strengths and weaknesses of this system?

Six people between the ages of 45 and 70 were interviewed, all of whom were linked to the Maestrat MTAS. The mean age of the interviewees was 58, which may seem very high, but it corresponds to the profile of the average farmer in the area. All of them were men: no women with an appropriate profile were found to be interviewed. The interviewees came from three different villages: La Jana, Traiguera, and Canet lo Roig. In these three municipalities, olive tree crops and millenary olive trees are remarkably widespread.

Two representatives of cooperative agro-industrial systems, two private producers, and two farmers who were members of a farmers' union were interviewed. The first four interviewees were selected as representatives of the two main systems of production in Maestrat (cooperatives and private production) and the last two as members of a farmers' union that has fought to preserve the Maestrat cultural landscape. In addition, one of the private producers interviewed is also a local development councillor in a Maestrat town council. Two of the interviewees were educated at college level, one of them had studied at an agricultural school, and the other three were educated at high school level.

The interview period was October 2022. The interview data collection method was audio recording. The interviews were conducted in person and in their respective villages. The language used was Catalan/Valencian, which is the language commonly used by all the interviewees [31].

As for the method for the assessment of landscape and heritage quality [30], it is adapted to the particularities that define agrifood systems. Different studies concerning the evaluation of landscapes were consulted for elaborating this method [32–34]. UNESCO criteria for inclusion in the World Heritage List were also analysed, as well as those considered by the FAO to select GIAHSs [30].

This assessment method, which can be seen in Table 1, is based on three categories of values: intrinsic (1), heritage (2), and potential and viability (3). Intrinsic values are cultural constructs that are projected onto inanimate objects and places; they refer to the inherent characteristics of the landscape itself, which are subjective and can mutate over time [5], and refer to values of representativeness, authenticity, ecological integrity, and visibility and visual quality. Heritage values include exogenous attributes—cultural or environmental—that influence and enhance the inherent characteristics of the landscape; they include historical, social, symbolic, identity, artistic, informative, and scientific values. Potential and viability values refer to the capacity of a landscape to be enhanced through the assessment of its production, security and quality of food, its awareness among social stakeholders, its participation and integration in local communities, its socio-economic profitability, its vulnerability, and its accessibility.

**Table 1.** Method for evaluating MTAS landscape and heritage quality.

| Intrinsic Values | | Heritage Values | | Potential and Viability Values | |
|---|---|---|---|---|---|
| | Representativeness | | Historic | | Food production, security, and quality |
| | Authenticity | | Social | | Awareness among social stakeholders |
| | Ecological integrity | | Symbolic/ Identity | | Participation and integration of local communities |
| | Visibility and visual quality | | Artistic | | Socio-economic profitability |
| | | | Informative/Scientific | | Vulnerability |
| | | | | | Accessibility |

Sources: Mayordomo-Maya and Hermosilla-Pla [30].

The Maestrat olive growing MTAS has a territorial continuity including ten municipalities, chosen following statistical criteria based on olive groves covering more than 40% of the cultivated area in each municipality. In some municipalities, the proportion of olive groves is much higher: Canet lo Roig (93%), La Jana (87%), Rossell (84%), Sant Mateu (76%), Traiguera and Xert (65%), La Salzadella (64%), and Tírig (61%). In Sant Rafel del Riu (44%) and Cervera del Maestrat (41%), the percentage is lower (Figure 1) [35].

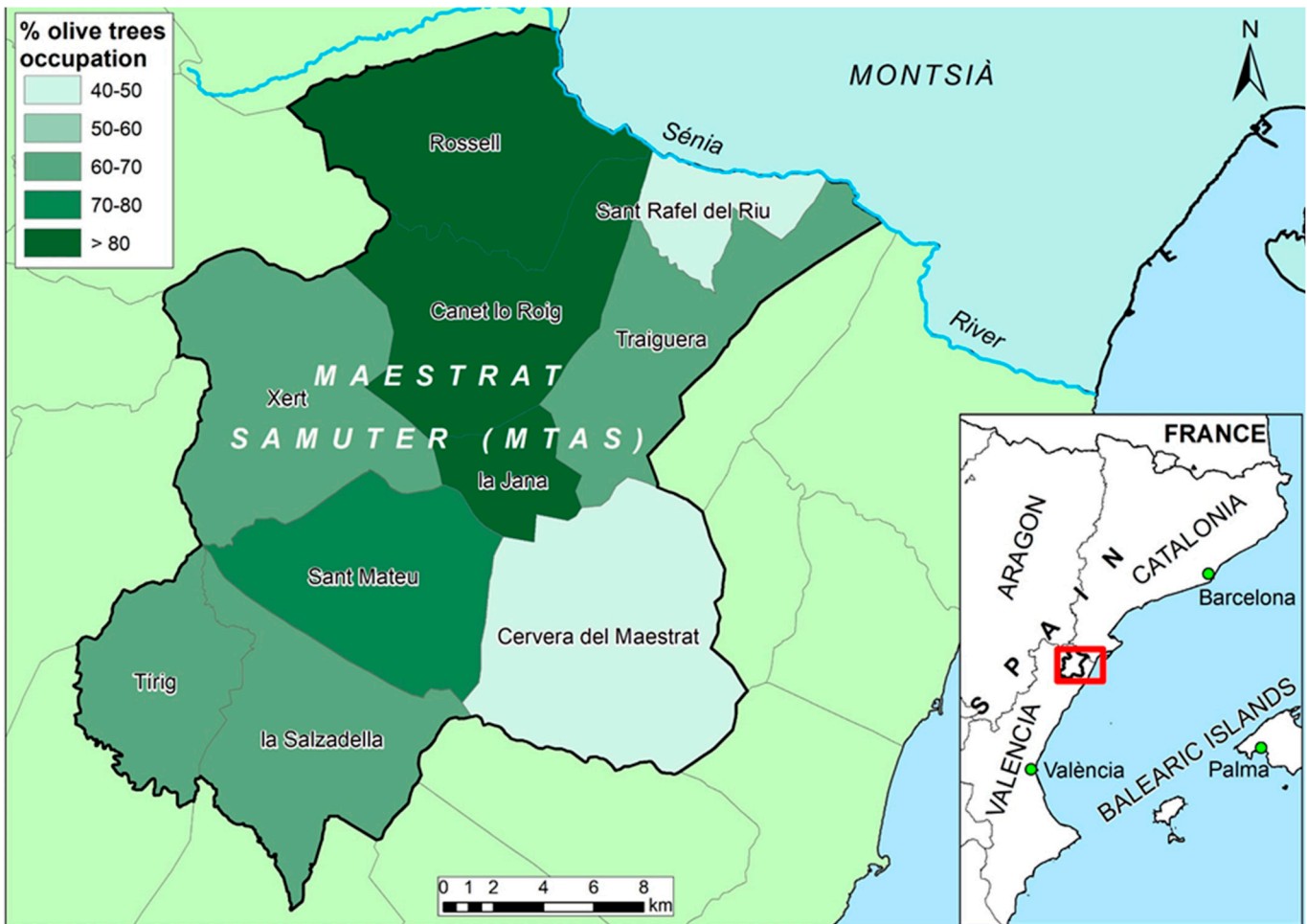

**Figure 1.** Municipalities included in the Maestrat MTAS. Source: Statistical Portal of Generalitat Valenciana [35].

Almost all the olive cultivation in the Maestrat MTAS is rain-fed, and indigenous olive varieties predominate (such as farga, morruda, cuquello, or canetera). This olive growing Valencian area and the adjoining Catalan area, separated by the River Sénia, have the highest density of thousand-year-old olive trees in the world (Figure 2). For this reason, they were recognized by the FAO in 2018 as GIAHS (Globally Important Agricultural Heritage Systems) under the name of Millennial Olive Trees Agricultural System of the Sénia Territory. This recognition was promoted by the *Mancomunitat de la Taula del Sénia*, a municipal commonwealth including most of the municipalities–although not all—of the Maestrat MTAS and others from the neighbouring territories of València, Aragón, and Catalonia (Figure 2). The criteria for delimiting the olive growing GIAHS of the Sénia Territory (based on distance from the River Sénia) are different from those of the Maestrat olive growing MTAS (based on more than 40% of olive trees over the total agricultural surface area), but, given the relevance of GIAHS recognition, it has been considered relevant to show the overlapping map of both territories (Figure 2).

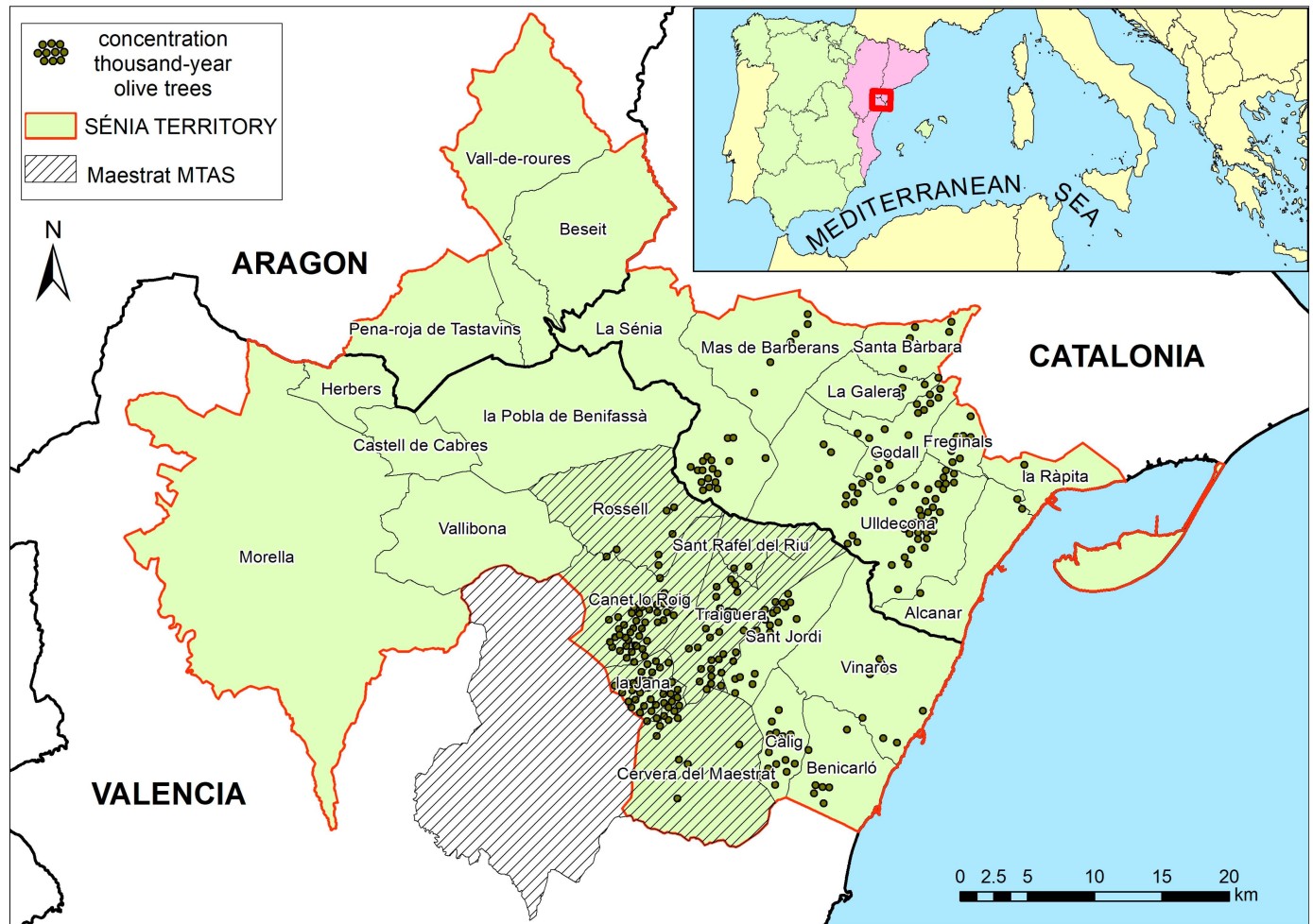

**Figure 2.** Millennial Olive Trees Agricultural System of the Sénia Territory (GIAHS). Source: Municipal Commonwealth Taula del Sénia [16,36].

## 3. Results

The method for analysing landscape and heritage quality described in the methodology [30] assesses the intrinsic values, the heritage values, and the potential and viability values of a given territory. This assessment was conducted using the materials from interviews, literature studies, and fieldwork.

Interviews were a crucial qualitative tool to assessing the results on the Maestrat olive growing MTAS. It can be seen how the interviewees describe the same issues from different perspectives, which helps to better understand their particular interests. For the cooperative leaders, cooperatives are a way of subsistence that is essential for the survival of local agriculture; for the private producers, cooperatives are a production system that prevents innovation; for the farmer's unionists, cooperatives are a plausible system, but they are currently too small, and a large regional cooperative would be advisable. All of them agreed on the problems of olive growing associated with the climate and soil. Cooperative leaders were not very concerned with preserving the landscape, although they recognized that the ancient olive trees are a qualitative asset for selling their products. Private producers and trade unionists defend the landscape's values. Cooperative members also claimed that the poor quality of the olive oil is inevitable due to climate limitations, but private producers—aware of this constraint—innovate by adapting the harvest to climate conditions and improving the quality of the oil.

### 3.1. Intrinsic Values

3.1.1. Representativeness

The intrinsic values of this MTAS are, in terms of its representativeness, the agricultural predominance of olive groves since, in each municipality, this kind of crop surpasses 40% of the agricultural surface area, and, in some municipalities, such as La Jana, Canet, and Rossell, it exceeds 80%. Moreover, olive growing has continued to expand over the last half century, with the same indigenous olive varieties, to the detriment of vineyards and cereals. The total sum of hectares in our study area is close to 15,000 [35]. This olive growing landscape produces a small part of extra virgin and single-varietal oil, which is appreciated for its excellent quality; however, most of Maestrat's production is *lampante* oil. *Lampante* is a Spanish word—cf English word *lamp*, from Latin origin—since this kind of oil was of poor quality and mainly used for producing light in lamps. *Lampante* oil cannot be directly consumed: it must be refined through an agro-industrial process and is often sold as oil for canned vegetables or canned fish in extra-regional markets. The main value of this landscape as a whole comes from its immense olive groves with little altered plot and road structures and with traditional dry stone architecture, but, above all, the significance of the Maestrat MTAS landscape derives from its monumental thousand-year-old olive trees, which are the living testimony of an ancient cultural landscape dating back to, at least, the medieval Muslim period.

3.1.2. Authenticity

In terms of authenticity, the immense olive groves, in general, and the thousand-year-old olive trees, in particular, together with the traditional dry stone architecture and the survival of the traditional road and plot structures, represent a well-preserved landscape identity where the close link between human beings and nature is evident. Production practices have gradually evolved to be adapted to the new socio-economic logic: the manual work of pruning the olive tree branches and harvesting the olives from the tree or from the ground has been mechanized and adjusted to the current socio-economic requirements and to improving farmers' quality of life. No landscape recovery measures have been applied since crops have not been massively abandoned or degraded. In general, most farmers are mainly concerned about economic yields, but a number of them are also concerned about preserving the landscape, preventing the uncritical sale of millenary olive trees, and opposing the installation of large wind or solar farms that fragment and distort the landscape (Figure 3).

Figure 3 shows that, for the time being, large wind and solar power plant projects remain far from the Maestrat olive growing MTAS. However, on the eastern periphery, the one closest to the coast, there are several solar farm projects (in Sant Jordi and Càlig). Furthermore, in the southern Maestrat MTAS, in an area not included in the Sénia Territory GIAHS, there are some small solar farm projects in Sant Mateu, Tírig, and la Salzadella and a large one in Les Coves de Vinromà, outside the Maestrat MTAS but just bordering it. The southern MTAS area also has wind farm projects in Tírig and, outside but bordering the MTAS, in Alcalà de Xivert.

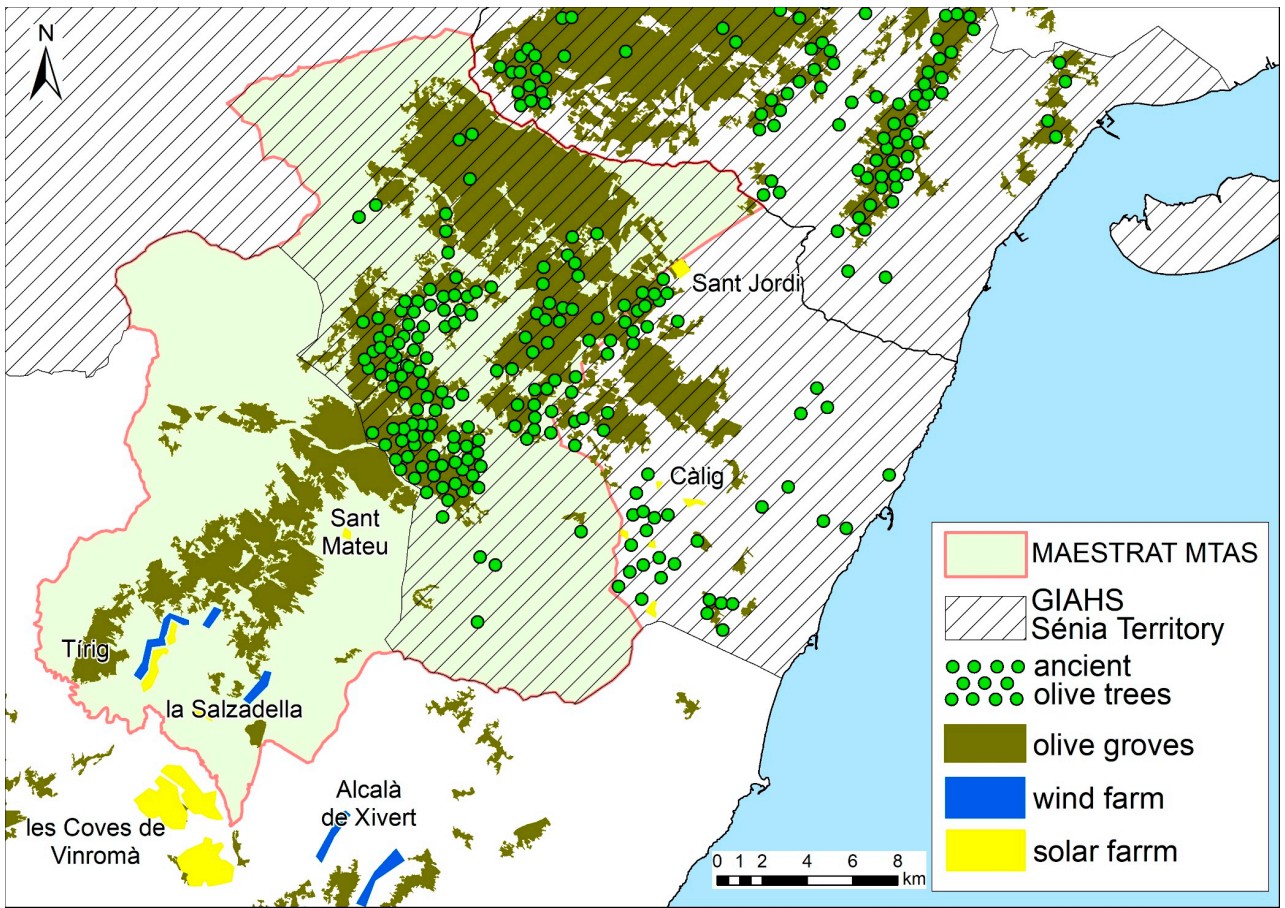

**Figure 3.** Wind farm and solar farm projects in the vicinity of the Maestrat MTAS. Sources: MCTS [36], VCI [37], and EEA [38].

The reasons why a large part of the Maestrat olive growing MTAS has been spared, for the time being, from the installation of renewable energy macro-projects is due to the high degree of protection that the area enjoys as more than half of the Maestrat MTAS is also part of the Sénia Territory GIAHS, where the concentration of thousand-year-old olive trees is larger. We say *for the time being* because this GIAHS protection does not imply that renewable energy plants will not be installed in the future, as has been seen in the very highly protected UNESCO Biosphere Reserve of Menorca (Spain), whose landscape is currently threatened by the installation of such projects [39].

If we leave the core of the GIAHS area, small renewable energy projects proliferate in the Maestrat MTAS itself, and outside but contiguous to it (Figure 3).

### 3.1.3. Ecological Integrity

As far as ecological integrity is concerned, avifauna is present in this area and acts as an ally for farmers in preventing pests and reducing the negative effects of the bactrocera oleae, also known as the olive fruit fly. The benefits for birds and olive trees are mutual [40], and it stops the use of pesticides to prevent the damage caused by the olive fruit fly [25]. Water pollution is not a pressing problem, and the indiscriminate use of chemicals is now more controlled than in the past thanks to some EU restrictive measures. There are contrasting differences between organic practices based on the use of natural processes and conventional practices, which do use chemical processes (such as fertilizers and pesticides) [41]. However, the Maestrat environment is not particularly degraded as the area is sparsely populated and not very industrialized. Organic farming, although still not widely used, has recently been promoted by the new EU CAP (2023–27) eco-schemes, which offer economic rewards for maintaining vegetation cover under the tree while it is in

winter dormancy or mechanical weeding techniques that regenerate the soil and combat erosion as opposed to the use of chemical products (Figure 4) [15].

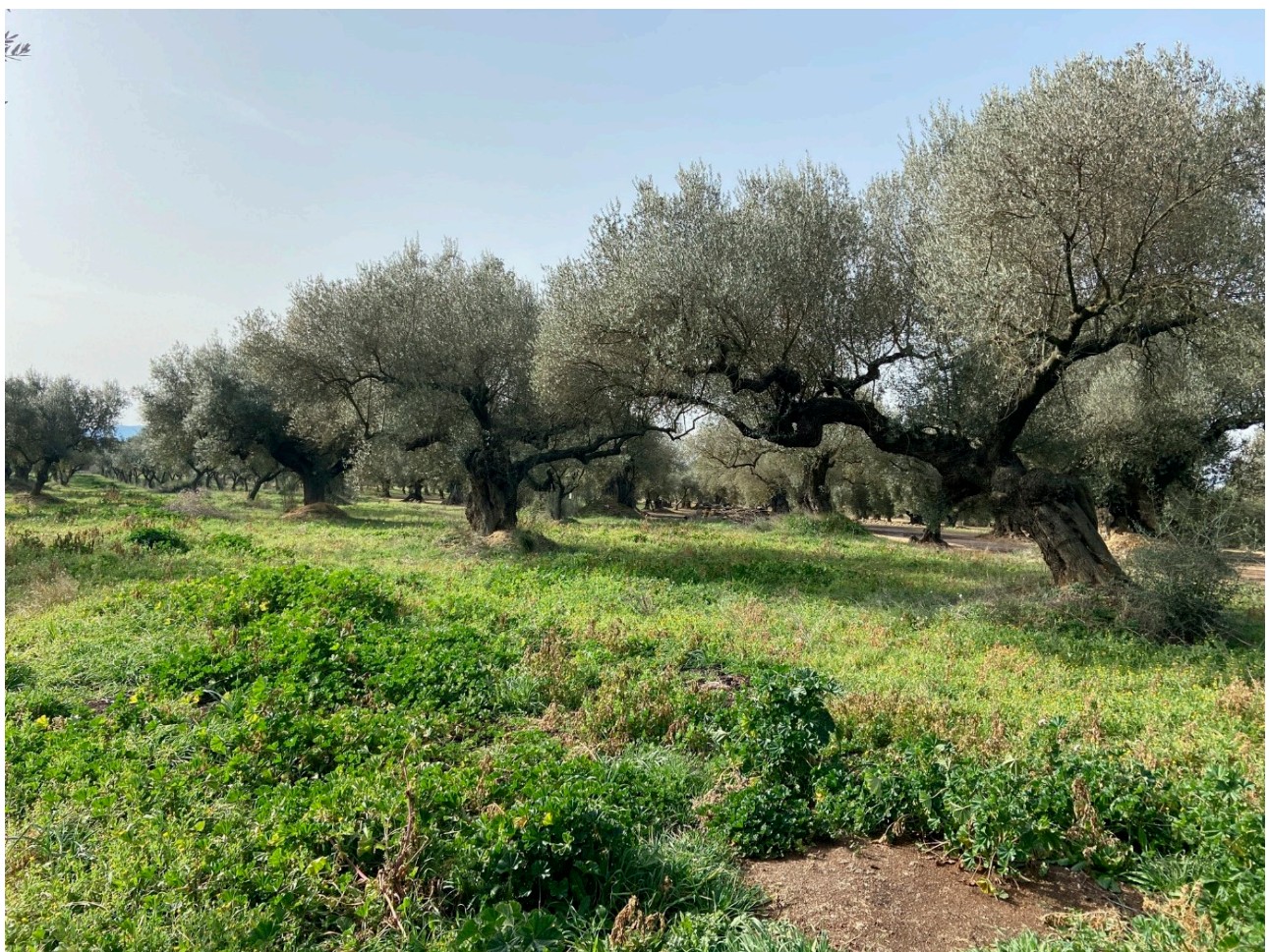

**Figure 4.** Organic practice: vegetation cover under olive trees in winter dormancy. Source: Àlex Vilanova Pla (La Jana, Maestrat).

As for ecosystem services, the current almost total monoculture of olive trees in the Maestrat MTAS offers such services to the non-agricultural population, for which farmers should be compensated by the political authorities for the maintenance of this non-urbanized vegetal landscape. Among these ecosystem services, it is important to mention food (olive oil), soil structure (by controlling erosion), absorption of $CO_2$ emissions (its biomass sequestering of atmospheric $CO_2$, which becomes part of the woody structures of plants, reduces atmospheric carbon dioxide, which accelerates climate change due to excess emissions), and hydrological services (by mitigating flood effects or recharging groundwater). The only local councillor interviewed expressed his helplessness and disappointment regarding how regional and national politicians do not support farmers for the maintenance of the landscape.

### 3.1.4. Visibility and Visual Quality

This refers to the breadth of the observable territory, visual connectivity with other territories, and visual range. These criteria help to describe the landscape from a scenic perspective [30]. In terms of visibility and visual quality, the view of the olive groves from the viewpoints is harmonious and organized and generates well-being in the observer. The main lookout points are the Mountains of Sant Pere (La Jana, Traiguera) and Vall d'Àngel (Sant Mateu) (Figure 5).

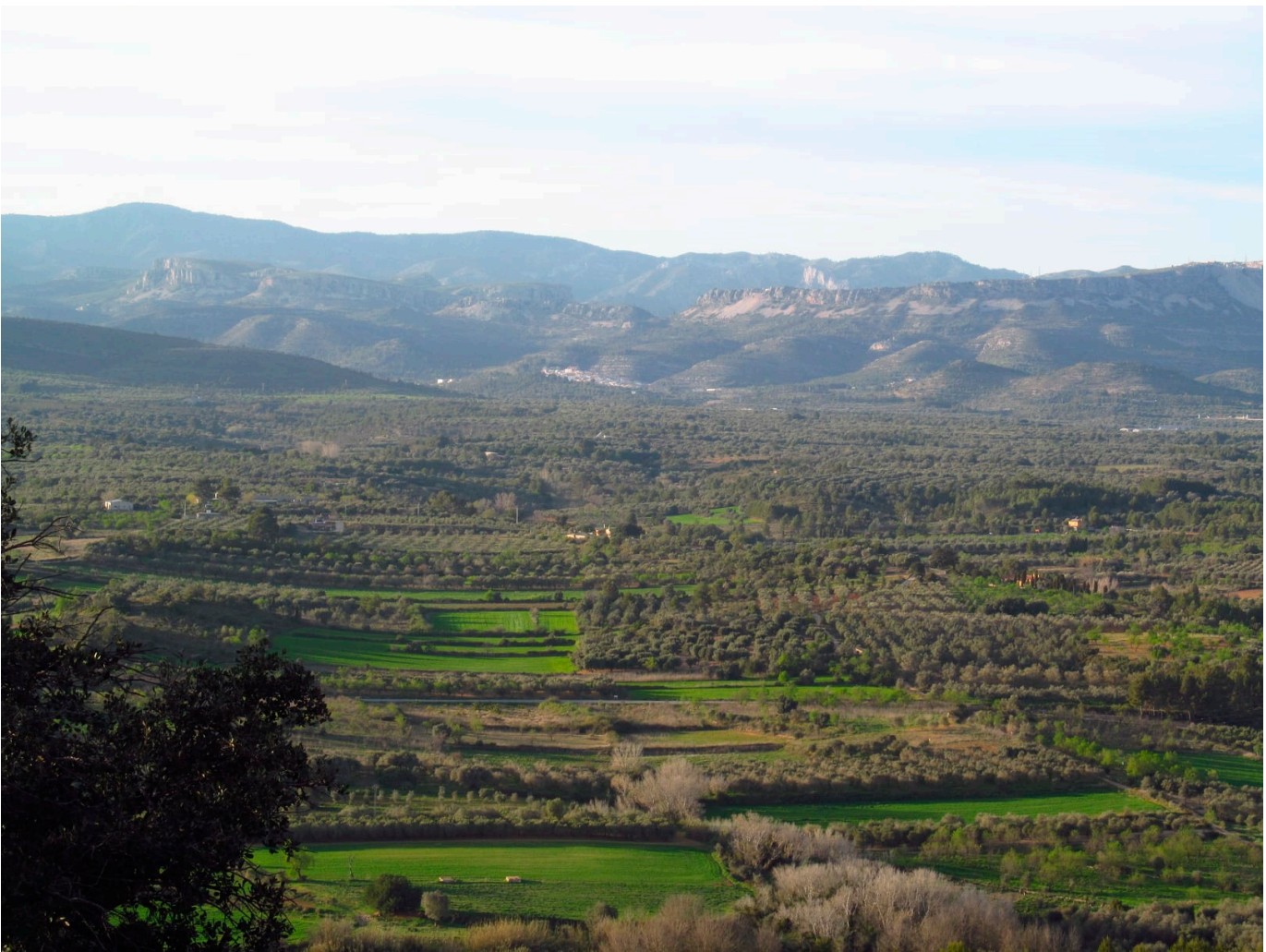

**Figure 5.** Viewpoint of Vall d'Àngel (Sant Mateu). Source: Antoni Martínez Bernat.

*3.2. Heritage Values*

3.2.1. Historic

The historic values in the Maestrat olive growing MTAS date back to the Iberians who, under Greek influence, learned to graft wild olive trees and turn them into olive trees for farming [42]. During the expansion of the Roman Empire, olive cultivation increased and methods of gardening, harvesting, grinding, and oil storage were improved. Agriculture before the Arabs was harmonious with the Mediterranean climate: the crops that make up its trilogy—olive trees, vines, and cereals—are merely an adaptation of species present in the Mediterranean forest [43]. With the arrival of the Arabs in Spain (seventh century), the range of crops was extended, without abandoning the classic Mediterranean trilogy of crops [44].

Christian settlers (thirteenth century) inherited and maintained this ancestral olive growing tradition. There are notable testimonies in Maestrat medieval documentation that refer to the significance of olive groves and oil mills [36]. It is worth mentioning that some historic oil mills are currently in ruins, but others have been renovated, such as the *Molí d'Oli* of Cervera del Maestrat, which was built in medieval times and worked until 1920. It was declared a Cultural Interest Asset by the Generalitat Valenciana (Regional Government of València) in 2007 and converted into an interpretation centre in 2018 (Figure 6).

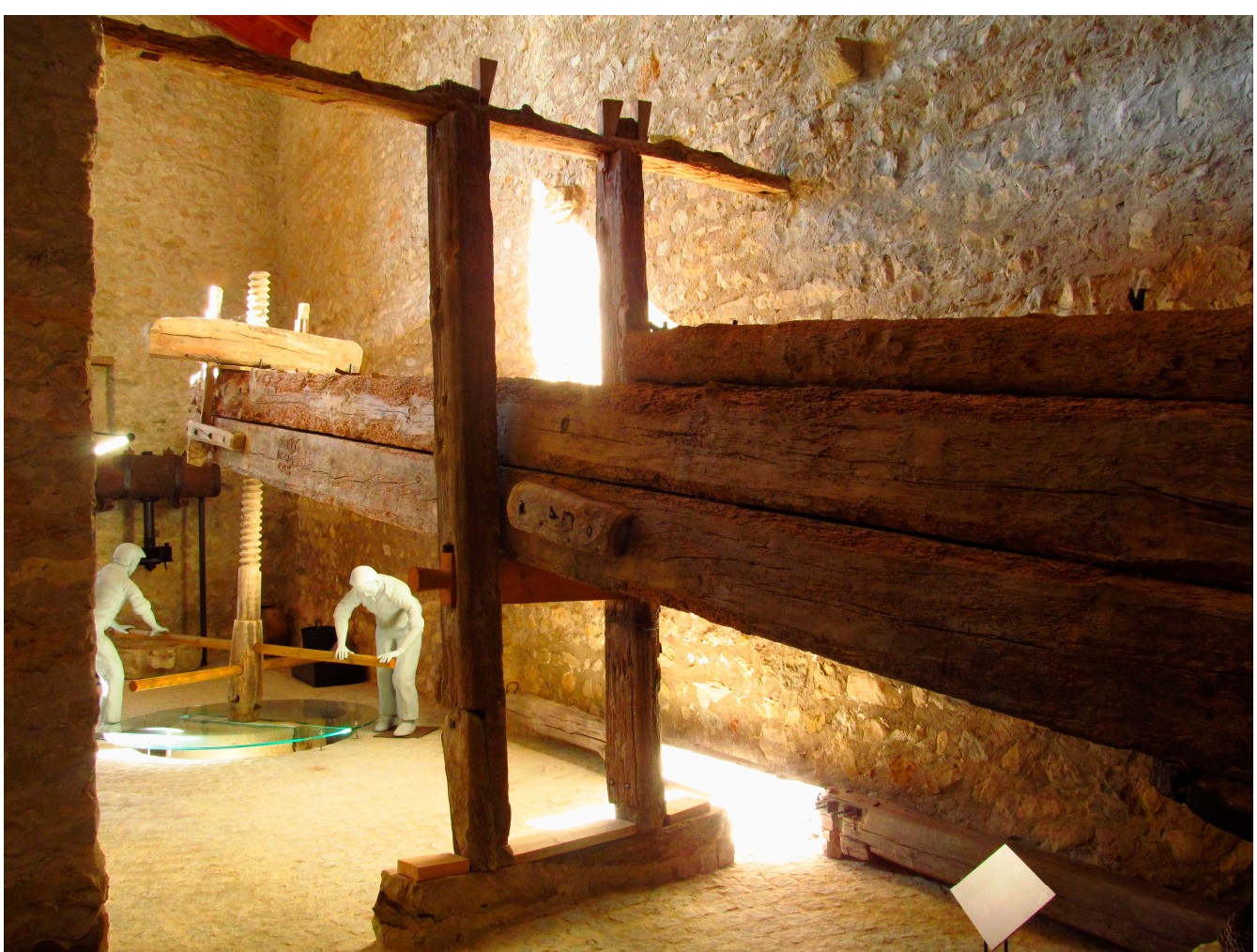

**Figure 6.** Oil mill of Cervera del Maestrat (interpretation centre). Source: Antoni Martínez Bernat.

During the second half of the twentieth century, olive trees ceased to coexist harmoniously with cereals and vines. The more lucrative olive trees replaced most cereals and vines and thus the Maestrat MTAS became nearly a monoculture. This process is not exclusive to Maestrat: it affected large rain-fed areas in Mediterranean Spain where, before their expansion, olive groves were a part of a diverse farming system, being just one more crop among a wide range of them [21].

### 3.2.2. Social

As far as the social context is concerned, the synergy between cooperatives and farmers in the territory should be mentioned (see Section 3.3.3). Cooperatives offer agricultural and banking services to farmers and machinery for both cooperative and private producers. Maestrat's farmers are generally elderly people who need cooperatives for their own subsistence.

### 3.2.3. Symbolic/Identity

This refers to the emotional ties and perceptions from local people as far as landscape is concerned. It considers symbolic bonds and identity perceptions [30]. The symbolic and identifying aspects of the Maestrat MTAS derive from its thousand-year-old olive trees, which have survived peoples, cultures, frosts, floods, and droughts. During the 1990s and up to 2006, some of these ancient trees disappeared from their environment due to speculation about their use in distant private gardens as mere ornamental objects, although the majority of them have survived thanks to an active group of defenders of the Maestrat

local landscape. Defenders of the millenary trees—such as the College of Pharmacists of the province of Castelló, as well as the Friends of the Olive Tree Association and the Valencian farmers' union *Unió Llauradora i Ramadera*—advocated for their protection and managed to push through a Valencian Law of Monumental Tree Heritage, unanimously approved by the Valencian Parliament [45]. Today, more than 2500 thousand-year-old olive trees in Maestrat (almost 5000 if we add up those in the neighbouring Catalan region of Montsià) are a living testament to this dynamic cultural landscape and its complex Mediterranean identity [46] (Figure 7).

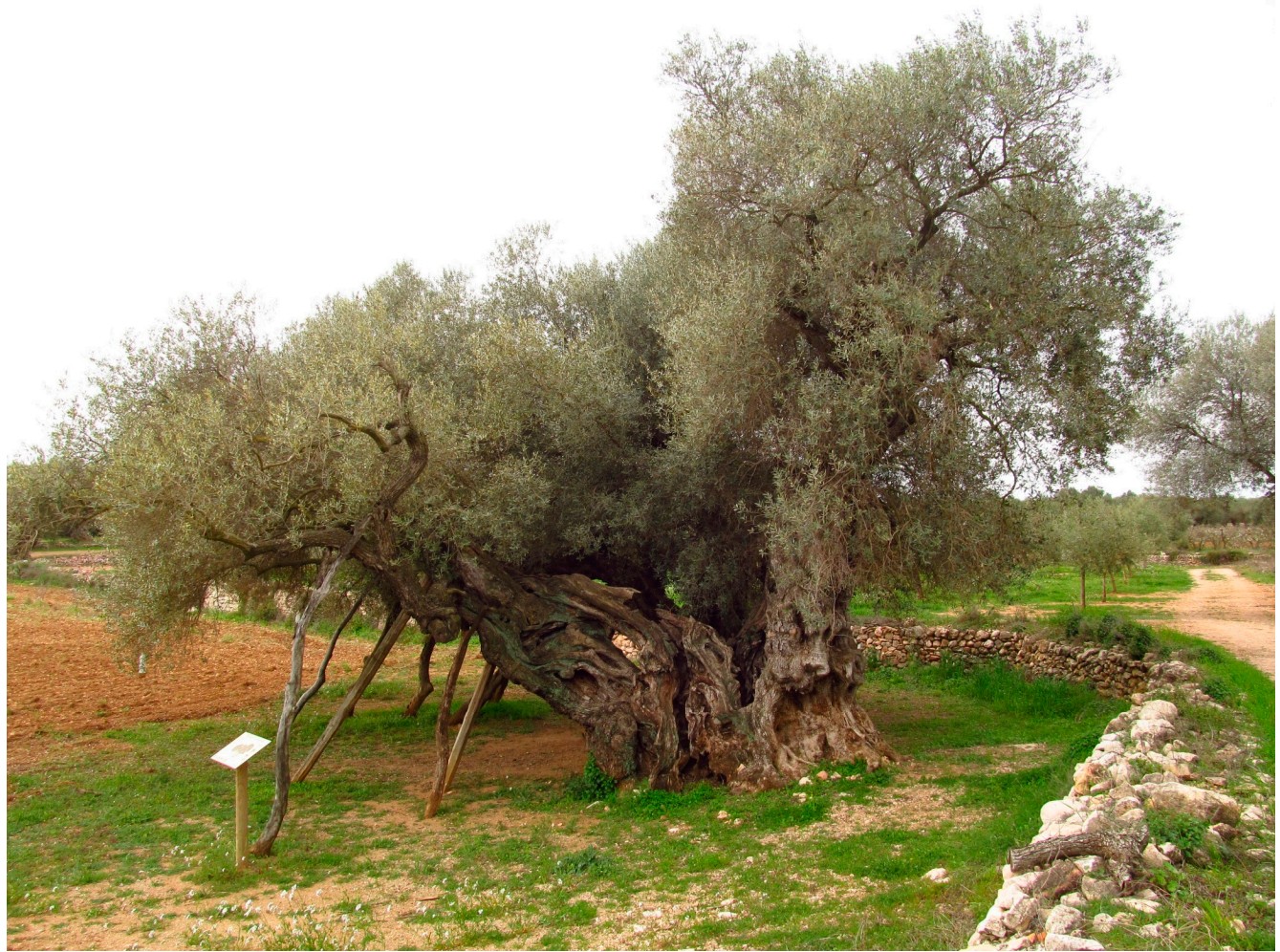

**Figure 7.** Thousand-year-old olive tree (Olivera de les Pitges, La Jana, Maestrat). Source: Antoni Martínez Bernat.

Maestrat olive trees, particularly the ancient ones, share their space with dry stone wells, huts, and walls. It should be noted that dry stone architecture, UNESCO World Heritage since 2018 [47], constitutes an outstanding cultural value and an environmental and economic resource since low dry stone walls (*marges* in the Catalan/Valencian language) between land plots help to fix the sloping soil in case of flooding (Figure 8). Moreover, dry stone huts (*barraques*) shelter the farmer in the event of storms, and dry stone wells are used to extract the scarce and precious water resources of the area. Dry stone, as a physical expression of human beings' ability to adapt to the environment, is an essential element that gives character to the Maestrat olive growing landscape (Figure 8).

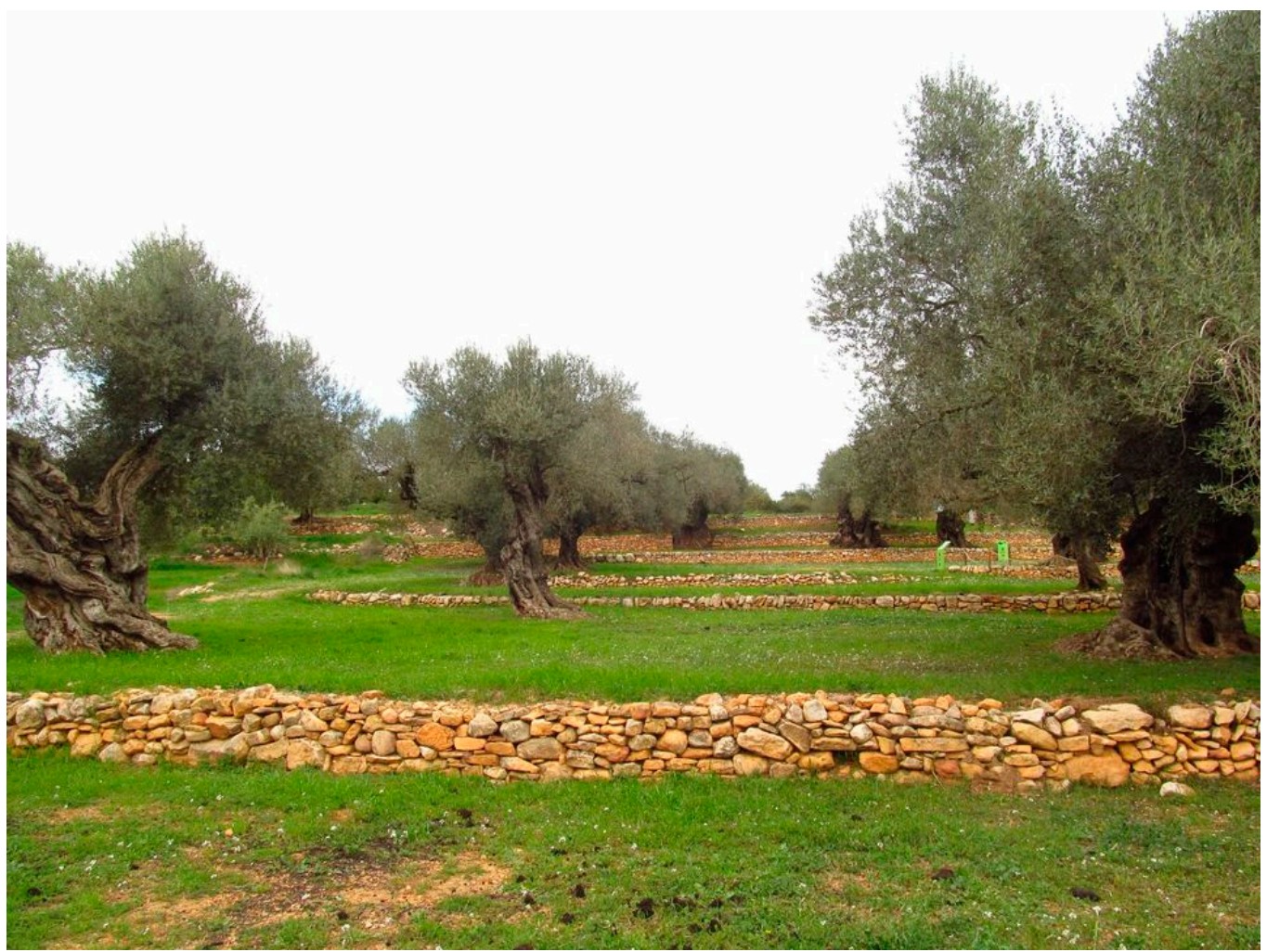

**Figure 8.** Low walls levelling the ground (La Jana, Maestrat). Source: Antoni Martínez Bernat.

### 3.2.4. Artistic

This considers artistic expressions linked to the landscape. Landscape can be represented through different artistic languages [30]. Perhaps the most notable artistic expression associated with this landscape is represented by the film *El Olivo*, directed by Icíar Bollaín (2016), in which a 20-year-old girl decides to go to Düsseldorf (Germany) to recover the thousand-year-old tree that her family sold. On the other hand, some activists in favour of ancient olive trees mobilized the civil society of the region, including painters and writers who contributed to spreading the artistic side of this MTAS.

### 3.2.5. Informative/Scientific

Unlike other Spanish GIAHS, such as the one in Horta de València [30] or in La Axarquía in Málaga [48], no research article at the international level has been specifically written about the thousand-year-old olive trees of the Valencian Maestrat. This article aims to rectify this situation and highlight the value of this thousand-year-old cultural landscape in the international scientific community.

### 3.3. Potential and Viability Values

### 3.3.1. Food Production, Security, and Quality

Most of Maestrat's oil is of poor quality and needs to undergo an agro-industrial process of refining and *coupage* (blending) for its conversion into virgin oil and subsequent distribution in medium-distance markets, such as the canned vegetables industry in

Navarre (northern Spain). Environmental conditions determine the production of this low-quality oil since, in November and December, Atlantic winds blow violently through the Ebro valley and flow into the Mediterranean through several regions, including Maestrat. More than 80% of the olive harvest can fall to the ground due to these strong winds, and the farmers have to collect the harvest from the ground using mechanized spiked cultivators. The quality of the olives in these conditions does not enable the production of extra virgin olive oil. Local cooperatives play a fundamental role in the storage, agro-industrial refinement, and marketing of these fallen and deteriorated olives. Less than 20% of Maestrat's oil production is harvested from the tree early, before the arrival of the winds, and is treated and distributed as a single-varietal-quality extra virgin oil. Due to its early harvesting, this extra virgin oil has spicy and bitter nuances, which give it an original flavour that is highly appreciated at international fairs, which have recently awarded prizes to different brands of Maestrat oil (such as the *Oli de Sant Jordi* company, which positioned its *Radix Nostra Nana* oil as one of the 10 best oils in the world among oils of limited production) [49]. This type of high-quality oil is managed by small producers who often practice organic farming and sell their product through short distribution channels (normally inside the Valencian territory). In addition, both small farmers and cooperatives often produce a small part of extra virgin oil from ancient olive trees, which is an attraction to arouse buyer interest in quality production linked to an extraordinary and unique landscape.

### 3.3.2. Awareness among Social Stakeholders

Among the social agents, the cooperative leaders manage the majority of poor-quality oil production following the agro-industrial method and its subsequent distribution in more or less distant markets, while the small producers of quality extra virgin oil follow more environmentally plausible methods (good practices) and sell in closer markets.

### 3.3.3. Participation and Integration of Local Communities

It is worth highlighting the synergy that occurs between the cooperatives and the residents in Maestrat's villages. There is a cooperative in almost every municipality, and some offer agricultural and banking services (credit lines). In addition, private producers mill their harvest using the cooperative's own machinery. Maestrat's villages depending on olive cultivation are small, ageing, and in the process of depopulation, so they must necessarily resort to cooperative formulas for their own subsistence.

### 3.3.4. Socio-Economic Profitability

Cooperative production often yields mediocre results as prices are not set by the cooperative membership but by external distributors. Few farmers can survive on their income from olive growing: they have to manage at least 35 to 40 hectares and often have to supplement their income with livestock income from intensive farms adjacent to the olive growing property and, above all, with CAP aid. Without EU help, most producers would stop caring for their land and the landscape would disappear. Unlike the fertile Guadalquivir Valley (the world's largest olive oil producer), the dry land of Maestrat is chalky and poor and the climate conditions make the quality of the product precarious if the olives are not harvested before the winds come and knock them to the ground. On the other hand, as said above, in line with the MTAS criteria, some non-cooperative producers overcome the environmental difficulties by harvesting early (before the fruit is expected to fall) and achieve a notable olive growing quality, complemented by good ecological practices. This added product value makes it possible to better control sale price, and it is normally sold at a local (regional) level. It must be said there is an appellation of origin for olive oil from the Region of València, but the Maestrat MTAS does not use it since it has its own label of quality linked to the thousand-year-old olive trees [50]. Around this label of quality, there is economic diversification linked to olive oil and olive tree tourism [51–53], with synergies between restaurants, shops, museums, and routes visiting traditional olive groves and their dry stone architecture [36].

### 3.3.5. Vulnerability

Vulnerability derives from low economic income, as explained in the section above. Furthermore, the protection of the landscape through the GIAHS does not permit the plundering of millenary olive trees anymore and protects the area from the installation of large renewable energy plants. However, farmers are not compensated for the lack of expected income due to the growing rural–urban interaction [54]. Moreover, Maestrat olive farmers feel aggrieved since the only public aid that they receive, coming from the CAP, is three times less than that received by Andalusian olive farmers as a result of a negotiation between Spain and the EU that goes back more than two decades and which urgently needs to be reviewed.

These aspects make Maestrat farmers vulnerable, and the younger ones will hardly maintain their olive grove system if they have no expectation of economic improvement. The ageing of farmers and the lack of generational replacement is, therefore, the most vulnerable aspect of this farming system.

### 3.3.6. Accessibility

The Maestrat olive growing MTAS has a notable advantage over other agricultural areas in the process of depopulation as it is only 20–30 kilometres away from a conurbation formed by towns such as Vinaròs, Benicarló, and Peníscola (a remarkable heritage landmark and beach resort). In this urban area, there are plausible non-agricultural employment alternatives for olive farmers, who can work there and at the same time continue to live in their villages and earn a supplementary income from their olive production while maintaining the extraordinary olive tree landscape of Maestrat. On the other hand, the closeness of this conurbation and its labour market is also a drawback since it simplifies the abandonment of olive farms, especially by young people, who prefer to be employed in this urban area.

## 4. Discussion

According to the results, the Maestrat olive growing system can be considered to only partially be an MTAS. As far as its production is concerned, it is mostly linked to hegemonic agro-industrial systems that produce food without differentiated territorial characteristics. This agro-industrial olive oil production is managed through cooperatives that enable the survival of dozens of farming families by managing the cultivation, harvesting, and sale of the olive production of farmers who are either too old to aspire to better economic results or who have another main source of income and only use their olive fields as a supplementary source of income.

Regarding cooperatives, it must be said that new global consumption patterns and increasing competition push farmers and food producers to innovate in order to look for more efficient production and distribution structures. In recent decades, the farming and food industries have shown growing collaboration since more coordination may lead to better efficiency in production and distribution. Changes in the food market raise the question of whether cooperatives are still efficient organizations for processing and marketing agricultural products [55,56].

In the case of the Maestrat MTAS, cooperatives do not work as an innovative business for improving the farmers' income but just as a survival tool to ensure a minimum income—supplemented by CAP aid—which at least allows farmers to maintain their farms. In Maestrat, cooperatives represent entrepreneurial immobility, but at least they guarantee the survival of the landscape and, in this sense, contribute to reinforcing Maestrat's olive agro-system as an MTAS.

There have been some efforts in the past to create a large olive oil cooperative that would bring together all the small municipal cooperatives in the region. However, the weight of municipal politics prevented the establishment of this large regional cooperative, which could supply a sufficient quantity of oil to be able to negotiate prices with the distribution chains. Therefore, the role of municipal cooperatives in Maestrat, although



crucial in maintaining the landscape and guaranteeing farmers a minimum income, is insufficient as a catalyst of this agrarian system.

Only a small percentage of Maestrat's oil production can be considered high-quality. This production, where the olives have been harvested early, has been recognized and awarded, in some cases, for its quality. Currently, this production of quality extra virgin olive oil is still limited, but new producers can be added if pioneer producers manage to significantly improve their income through qualitative improvement, complemented by good agricultural practices. The latter are stimulated by the new CAP policy (2023–2027), which encourages organic farming practices and landscape protection [57].

In this way, we find a dual model of oil production in Maestrat: (1) the cooperative survival model, followed by the most farmers (especially the older ones and those with land as a secondary job), which allows for the precarious survival of farming, although it ensures, for the time being, the maintenance of the landscape; and (2) the model of private producers, followed by some young farmers, who live mainly on the land and opt for quality oil with a prestigious brand (some of whom have recently won international awards), good agricultural practices (in line with the new CAP landscape protection requirements), and a regional proximity market allowing food sovereignty. Closeness to the market is crucial because it mitigates climate change effects caused by the long-distance transport of food and allows food sovereignty, which is critical in moments of food distribution restrictions such as those recently experienced, caused by COVID-19 or the Russian invasion of Ukraine [54].

Both model one and model two permit, for the time being, the more or less precarious survival of the olive growing cultural landscape of the Valencian Maestrat. With regard to this cultural landscape, it must be said that the ecosystem services derived from it benefit people by providing food, maintenance of soil structure, hydrological services, and sequestering atmospheric $CO_2$. The ecological sustainability linked to ecosystem services has the potential to drive a paradigmatic change in land use managing and planning through political–economic decisions [58,59], such as those initiated by the EU with its new CAP (2023–27) eco-schemes. This provides stronger incentives for climate-friendly and environmentally friendly farming practices (such as organic farming) as tools for environmental care and landscape protection [60].

Beyond the profits derived from its ecosystem services, we must also refer to the benefits derived from the culture of this ancient landscape. Although landscape values are subjective and mutate over time [61–64], based on the current valuation criteria, there is a good degree of unanimity in considering the thousand-year-old olive trees of the Valencian Maestrat as an extraordinary landscape that must be preserved, as evidenced by the GIAHS recognition that it possesses. It must be said that, because of the actions of a few courageous farmers and agricultural trade unionists with a strategic and environmental vision for the future, the plundering of millenary olive trees was stopped after the approval of the Valencian Law of Monumental Tree Heritage in 2006. Later, in 2018, due to the political initiative of the Municipal Commonwealth *Taula del Sénia*, the Sénia territory (including most of the olive growing Maestrat system) was declared a GIAHS for its agricultural resilience and its ancient cultural landscape. These protective and conservationist measures have prevented the installation of disruptive renewable energy plants in the core area (the one with the largest number of thousand-year-old olive trees) and thus have improved the preservation of this landscape and its ecosystem services.

Nevertheless, further policies should now be aimed at protecting the weakest link in the chain: the farmers. They feel aggrieved because they do not receive any governmental aid (beyond CAP); they cannot sell more ancient olive trees and improve their income and neither will they benefit from solar or wind farms and profit from the land leasing. In contrast, they are responsible for maintaining this valued cultural landscape so that people (farmers or not) can enjoy its cultural and ecosystem services. Moreover, as far as CAP aid is concerned, Maestrat olive farmers feel that they are discriminated against when comparing their CAP aid with that received by Andalusian olive growers since the latter obtain much more aid per hectare because of a negotiation between the EU and Spain

that led to different payments depending on the olive growing region, with Andalusia benefiting from it to the detriment of the eastern Mediterranean regions [65]. Valencian farmers receive around EUR 250 per ha, while Andalusian farmers can receive up to three times more. This distorts the market because Andalusian oil is sold more cheaply.

For all the above-mentioned reasons, Maestrat farmers—and especially the younger ones—will hardly preserve their olive grove system if they do not see any economic prospects to motivate them to stay. This lack of motivation to continue working the land is a common feature of Mediterranean agriculture. It is argued that farming is unattractive for young people since it entails not only low income but also climatic risks, uncertain working hours, fluctuating selling prices, living in isolated areas, and a lack of public incentives [66]. The growing importance of cities as a labour market is a challenge as it facilitates the abandonment of farms by young people to look for work in nearby urban areas [67], as is the case of the Vinaròs–Benicarló–Peníscola conurbation in Maestrat, only 20 to 30 min away from the Maestrat MTAS.

## 5. Conclusions

In spite of the precariousness of its farmers, it can be said that the Maestrat olive growing landscape responds to MTAS criteria as regards its identity aspects based on a unique landscape with character. Cooperative agro-industrial production has hardly altered the plot or road structures, nor has it eliminated the traditional rural architecture of dry stone, nor transformed the historical rain-fed crops into irrigated (water-wasting) harvests. Moreover, the ecosystem services are remarkable, but the farmers who maintain them are not sufficiently compensated for them. Furthermore, legal protection, through a law protecting monumental trees (especially, thousand-year-old olive trees), has contributed to stopping the indiscriminate sale of ancient trees and preventing the installation of solar and wind farms. Additionally, publicity campaigns enhancing the value of this absolutely exceptional and ancient landscape have added prestige to Maestrat extra virgin olive oil. This cultural landscape is a tourist resource since, every year, thousands of tourists visit its thousand-year-old trees, not only from the neighbouring beaches of Peníscola but also from further afield.

On the other hand, much remains to be done in terms of quality oil production. Only a few farmers have decided to produce quality oil by harvesting the olives before the winter wind blows them to the ground and take advantage of the worldwide prestige offered by GIAHS landscape protection of ancient olive trees. These innovative farmers comply with the Sustainable Development Goals of MTASs, such as proximity between production and consumption, environmental sustainability, family farming, participatory governance, territorial attachment through local products, and landscape preservation, resulting in sustainable cultural tourism that generates employment and consumes local production. In contrast, most production is carried out through cooperatives and does not fulfill all these SDGs (it sells to medium-distance markets if the distribution channel dictates so, and produces poor-quality oil), but at least it does allow the precarious survival of family farming, which maintains a millenary landscape that offers extraordinary ecosystem and cultural services.

**Author Contributions:** Conceptualization, J.H.-P. and J.C.M.-T.; methodology, J.C.M.-T. and J.H.-P.; validation, J.C.M.-T. and J.H.-P.; formal analysis, J.C.M.-T. and J.H.-P.; investigation, J.C.M.-T. and J.H.-P.; resources, J.H.-P. and J.C.M.-T.; data curation, J.C.M.-T.; writing—original draft preparation, J.C.M.-T. and J.H.-P.; writing—review and editing, J.C.M.-T.; visualization, J.H.-P. and J.C.M.-T.; supervision, J.H.-P. and J.C.M.-T.; project administration, J.H.-P.; funding acquisition, J.H.-P. All authors have read and agreed to the published version of the manuscript.

**Funding:** This research was funded by Spanish Ministry of Science and Innovation. Project: Multifunctional and territorialized agrifood systems in Spain. Methodology to evaluate landscape and heritage, with cases in València. Grant PID2019-105711RB-C66 funded by MCIN/AEI/10.13039/501100011033.

**Institutional Review Board Statement:** Not applicable.

**Informed Consent Statement:** Not applicable.

**Data Availability Statement:** Not applicable.

**Conflicts of Interest:** The authors declare no conflict of interest. The funders had no role in the design of the study; in the collection, analyses, or interpretation of data; in the writing of the manuscript; or in the decision to publish the results.

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
