# Peer review of "The Precarious Survival of an Ancient Cultural Landscape: The Thousand-Year-Old Olive Trees of the Valencian Maestrat (Spain)"

_land, doi:10.3390/land12071331_

Round 1

Reviewer 1 Report

Review of land-2432135-peer-review-v1

The precarious survival of an ancient cultural landscape: the  thousand-year-old olive trees of the Valencian Maestrat (Spain

The paper has the potential to provide and interesting insight into the contemporary use and management of cultural landscapes in the olive producing areas. Unfortunately, it is methodologically flawed and needs a thorough rethink. Below please find some comments to aid the authors in their revision

The introduction needs to have  a much stronger foundation in cultural landscape theory…

METHODOLOGY

How were the interviewees chosen/selected? This needs to be explained. Why only two per cell in the sampling frame?  In line 137 you query “politicians’ awareness of it” as a discussion topic. Why did you not also include local politicians in the interviews?  Also, what about local government policy makers? The selection process and rationale for the selection of interviewees needs to be much better explained and justified.

Also, the epistemology of the views of the interviewees must be considered.

Line 145 ff       The methodology uses the term of ‘intrinsic’ values but its discussion makes it very clear that all values are in fact not independent of context but are cultural constructs and projections of community and decision makers …” representativeness” is a heritage related construct that is influenced by the valuer’s socialisation, so are the constructs of ‘authenticity’ and ‘visual quality’.  These values are subjective , not intrinsic, and are also mutable qualities. The authors may wish to consider the effects intergenerational change … see Spennemann, D. H. R. (2022). The Shifting Baseline Syndrome and Generational Amnesia in Heritage Studies. Heritage, 5(3), 2007–2027.

Even the construct of ‘ecological integrity” is a projected value construct that relies on the observation of  the present without considering historic trajectories and the effects of the shifting baseline syndrome …see Pauly, D. (1995). Anecdotes and the shifting baseline syndrome of fisheries. Trends in Ecology & Evolution, 10(10), 430  and Vera, F. (2010). The shifting baseline syndrome in restoration ecology. In Restoration and history (pp. 116-128): Routledge).

Line 147 ff       ‘Heritage values” again they are mutable qualities. So who participated in the interviews, to which generation did they belong to? What about the motivations to participate. It may be worth to consider the framework discussed in this paper and then address this as limitations if such an assessment was not carried out (which  it seems it was not): Spennemann, D. H. R. (2023). The usefulness of the Johari Window for the Cultural Heritage Planning Process. Heritage, 6(1), 724-741

As it stands, the methodology appears rather weak. It is also of concern that nowehere in the paper the authors address limitations to their study  and approach.

RESULTS 

Line 190ff, Representativeness

How was that assessed? So this is the pure percentage of cover? Many new groves seen to have been planted. Do they all have the same “indigenous olive varieties”? This is not clear. And if they don’t where does ‘Representativeness’ come from?

Line 203, now we are introducing dry stone structures.  As ‘Representativeness’’?

Line 205 “monumental thousand-year-old olive trees” what has that got to do with  “Representativeness’’ This is very much unclear.

Where are the views of the interviewees discussed here ?

Line 207ff Authenticity

“the man-212 ual work of pruning the olive tree branches and harvesting the olives from the tree or from 213 the ground has been mechanized, and adjusted to the current socio-economic require-214 ments and to improving farmers’ quality of life.”  SO, what is authentic here? The whole concept of ‘Authenticity is rubbery and not defined in the paper

Line 244 ff Ecological Integrity

“As far as ecological integrity is concerned, mention should be made of indigenous 244 olive varieties such as farga, morruda, cuquello or canetera”

This is not ‘ecological integrity’  where is the ecology discussed here? Thee are ample papers on the ecology of Spanish olive orchards (such as Rey, Pedro J. "Preserving frugivorous birds in agro‐ecosystems: lessons from Spanish olive orchards." Journal of Applied Ecology 48, no. 1 (2011): 228-237. and Infante-Amate, J., 2012. The ecology and history of the Mediterranean olive grove: the Spanish great expansion, 1750-2000. Rural History, 23(2), pp.161-184.) but I suspect this not the ecology that the authors have in mind…

Line 255ff Visibility and visual quality

This section is methodologically unsound

Line 196ff Social

This section is very weak . Whee is the social value here? Read Jones, S. (2017). Wrestling with the Social Value of Heritage: Problems, Dilemmas and Opportunities. Journal of Community Archaeology & Heritage, 4(1), 21-37

Line 303 Symbolic/Identity

This section is methodologically unsound

Line 332          Artistic

Best to drop this. Where is the relevance?

Line 340          Informative/Scientific

Scientific value in cultural heritage management has very specific meanings. Not hpw the authors interpret this…

I am stopping the review at this point…

MINOR ISSUES

Line 8  Not everyone reading the paper will know what “Valencian Maestrat” is…so you need to explain that…as you did in line 77

Line 45/46       consider citing:

Stephenson, Janet. "The Cultural Values Model: An integrated approach to values in landscapes." Landscape and urban planning 84, no. 2 (2008): 127-139.

Cosgrove, D. (1989). Geography is everywhere: culture and symbolism in human landscapes. Horizons in human geography, 118-135

Sentence ending line 61 needs references…

Sentence ending line 81 needs a reference…

Sentence ending line 83 needs a reference…

Sentence ending line 92 needs a reference…

Line 94…why “this must have been maintained” ?

Sentence ending line 98 needs a reference…

Line 100          There needs to be an academic reference that validates the dating and that cannot be the be that underpins Figure 2

Line 103 to 111            this assertion  needs to be referenced

Line 113 to 121            this assertion  needs to be referenced

Line 130          what is a ‘union farmer’ ? needs explanation

Figure 2           The total boundaries of the Valencia Maestrat should be shown, with the areas over 40% coverage with olive groves being shaded.

Line 166 what are what are they and what is the time depth here? This this assertion also needs to be referenced

LANGUAGE

The paper contains numerous infelicities in grammar and expression. It needs to be edited by a professional native-English speaking scientifically trained editor.

Line 99            what is an “empirical presence” ? Translation issue from the Spanish?

Line 113          ”aim of this article is to know if”

Line 123          “This communication”  this is designated an “Article” in line 1. What is it? A communication or an article…Surely the latter

Figure 2           “olive tree occupation”

ETHICS

Given that the study relies on  semi-structured interviews with representatives of cooperatives,  union farmers and private producers, there is a need that prior ethics approval has been obtained. The authors need to provide evidence to that effect from the Ethics Review Board of their University, 

The paper contains numerous infelicities in grammar and expression. It needs to be edited by a professional native-English speaking scientifically trained editor.

Author Response

Comments and Suggestions for Authors

Review of land-2432135-peer-review-v1

The precarious survival of an ancient cultural landscape: the  thousand-year-old olive trees of the Valencian Maestrat (Spain

The paper has the potential to provide and interesting insight into the contemporary use and management of cultural landscapes in the olive producing areas. Unfortunately, it is methodologically flawed and needs a thorough rethink. Below please find some comments to aid the authors in their revision

The introduction needs to have a much stronger foundation in cultural landscape theory…

We added this about cultural landscapes to the introduction:

A cultural landscape can be defined as a result of the physical environment being altered by social and economic forces over time [1]. Cultural landscapes have been given increasing attention with the influence of globalization and the notion of sustainable development. The progressive abandonment of farmlands with their associated landscapes has changed land structures and functions while accelerating the degradation of cultural landscapes and food production associated with them [2–3]. The purpose of this article is to analyze the Maestrat olive growing cultural landscape, which is a district to the north of the Region of Valencia (Spain) (Figures 1–3). 

METHODOLOGY

How were the interviewees chosen/selected? This needs to be explained.

We added this text to the article about the method on interview:

Significant qualitative information was obtained from the semi-structured interviews: six people between the ages of 45 and 70 were interviewed, all of whom were linked to the Maestrat MTAS. The mean age of the interviewees was 58, which may seem very high, but it corresponds to the profile of the average farmer in the area. All of them were men: no women with an appropriate profile were found to be interviewed. The interviewees came from three different villages: La Jana, Traiguera and Canet lo Roig. In these three municipalities, olive tree crops and millenary olive trees are remarkably widespread.

Two representatives of cooperative agro-industrial systems, two private producers and two farmers who were members of a farmers’ union were interviewed. The first four interviewees were selected as representatives of the two main systems of production in Maestrat (cooperatives and private production) and the last two as members of a farmers’ union that has fought to preserve the Maestrat cultural landscape. In addition, one of the private producers interviewed is also a local development councilor in a Maestrat town council. Two of the interviewees were educated at college level, one of them had studied at an agricultural school, and the other three were educated at high school level.

The interview period was October 2022. The interview data collection method was audio recording. The interviews were conducted in-person and in their respective villages. The language used was Catalan/Valencian, which is the language commonly used by all the interviewees [31]. The questions were agreed upon by the MTAS working group in the Department of Geography at the University of Valencia. The ten questions asked were (1) Do you think that this territory conforms to a MTAS? (2) If yes, who are the main actors in this MTAS? (3) Why does this system differ from others? (4) Does this system preserve its original landscape? (5) What traditional techniques and knowledge distinguish this system from others? (6) Is the current system environmentally sustainable? Is it socioeconomically sustainable? (7) What elements beyond farming make this system a multifunctional space? (8) Do the governmental authorities make investments to preserve this system? (9) Is there social identification with this system? (10) What are the main strengths and weaknesses of this system?

The interviews were crucial to understanding the olive growing context of Maestrat. It can be seen how the interviewees describe the same issues from different perspectives, which helps to better understand their particular interests. For the cooperative leaders, cooperatives are a way of subsistence that is essential for the survival of local agriculture; for the private producers, cooperatives are a production system that prevents innovation; for the farmer’s unionists, cooperatives are a plausible system but they are currently too small and a large regional cooperative would be advisable. All of them agreed on the problems of olive growing associated with the climate and soil. Cooperative leaders were not very concerned with preserving the landscape, although they recognized that the ancient olive trees are a qualitative asset for selling their products. Private producers and trade unionists defend the landscape’s values. Cooperative members also claimed that the poor quality of the olive oil is inevitable due to climate limitations, but private producers–aware of this constraint–innovate by adapting the harvest to climate conditions and improving the quality of the oil. All the answers were very significant and helped to better understand the general situation of olive growing in the Maestrat MTAS from a qualitative viewpoint.

Why only two per cell in the sampling frame?  

We consider that two is a sufficient sample for an agricultural area relatively small in size and number of farmers.

In line 137 you query “politicians’ awareness of it” as a discussion topic. Why did you not also include local politicians in the interviews?  Also, what about local government policy makers?

Actually, one of the interviewees is a council member (of rural development) in a municipality of the Maestrat. This remark has been added:

In addition, one of the private producers interviewed is also a local development councilor in a Maestrat town council.

The selection process and rationale for the selection of interviewees needs to be much better explained and justified.

This has been done, thanks to your plausible remarks.

Also, the epistemology of the views of the interviewees must be considered.

Two of the interviewees were educated at college level, one of them had studied at an agricultural school, and the other three were educated at high school level.

Line 145 ff       The methodology uses the term of ‘intrinsic’ values but its discussion makes it very clear that all values are in fact not independent of context but are cultural constructs and projections of community and decision makers …”

We have used the terminology used by Mayordomo-Maya, S.; Hermosilla-Pla, J. Evaluation of Landscape Quality in Valencia’s Agricultural Gardens—A Method Adapted to Multifunctional, Territorialized Agrifood Systems (MTAS). Land 2022 11(3), 398. https://doi.org/10.3390/land11030398

Intrinsic means “belonging to the essential nature or constitution of a thing”, and therefore it refers, by extension, to the particular characteristics of our study area.

representativeness” is a heritage related construct that is influenced by the valuer’s socialisation, so are the constructs of ‘authenticity’ and ‘visual quality’.  These values are subjective, not intrinsic, and are also mutable qualities.

The authors may wish to consider the effects intergenerational change … see Spennemann, D. H. R. (2022). The Shifting Baseline Syndrome and Generational Amnesia in Heritage Studies. Heritage, 5(3), 2007–2027.

The reference has been added

Even the construct of ‘ecological integrity” is a projected value construct that relies on the observation of  the present without considering historic trajectories and the effects of the shifting baseline syndrome …see Pauly, D. (1995). Anecdotes and the shifting baseline syndrome of fisheries. Trends in Ecology & Evolution, 10(10), 430  and Vera, F. (2010). The shifting baseline syndrome in restoration ecology. In Restoration and history (pp. 116-128): Routledge).

These references have been added

Line 147 ff       ‘Heritage values” again they are mutable qualities. So who participated in the interviews, to which generation did they belong to? What about the motivations to participate. It may be worth to consider the framework discussed in this paper and then address this as limitations if such an assessment was not carried out (which  it seems it was not): Spennemann, D. H. R. (2023). The usefulness of the Johari Window for the Cultural Heritage Planning Process. Heritage, 6(1), 724-741

We consider that the remarks mentioned by you must be taken into account for next future uses and improvement of this methodology, and the reference recommended by you has been added in the discussion.

As it stands, the methodology appears rather weak. It is also of concern that nowehere in the paper the authors address limitations to their study  and approach.

Thank you for your remarks: many ammends have been done to solve this thanks to you.

RESULTS 

Line 190ff, Representativeness

How was that assessed?

It was assessed by Statistical Portal of Generalitat Valenciana (Portal Estadístic de la Generalitat Valenciana). Territorial Data Bank. Estimates of cultivated areas by municipality (Banc de Dades Territorial. Estimacions de superfícies de cultiu per municipis), 2023. https://pegv.gva.es/va/

So this is the pure percentage of cover? Yes

Many new groves seen to have been planted. Do they all have the same “indigenous olive varieties”? This is not clear.

Yes, the same indigenous varieties.

Now it is not unclear anymore: “Moreover, olive growing has continued to expand over the last half century -with the same indigenous olive varieties-, to the detriment of vineyards and cereals.”

And if they don’t where does ‘Representativeness’ come from? But they do

Line 203, now we are introducing dry stone structures.  As ‘Representativeness’’?

Yes, dry-stone structures (recognized by UNESCO) are representative traits of the Maestrat olive growing system. 

Line 205 “monumental thousand-year-old olive trees” what has that got to do with  “Representativeness’’ This is very much unclear.

From our viewpoint, monumental thousand-year-old olive trees are the most representative trait of the Valencian Maestrat olive growing.  

Where are the views of the interviewees discussed here?

The method used in this article derives from the literature review, fieldwork, mapping, interviews and a method for assessing land-scape and heritage quality. Not every method is used for each subsection. But now, thanks to your remarks about methods, it has been added in this article that two of the interviewees are members of a farmers’ union that has fought to preserve the cultural landscape of Maestrat, so it is quite clear that both are defenders of this cultural landscape.

Line 207ff Authenticity

“the man-212 ual work of pruning the olive tree branches and harvesting the olives from the tree or from 213 the ground has been mechanized, and adjusted to the current socio-economic require-214 ments and to improving farmers’ quality of life.”  SO, what is authentic here? The whole concept of ‘Authenticity is rubbery and not defined in the paper

Manual processes have been modernized to improve the quality of life of farmers but local olive varieties, plot structures, or the landscape in general remain the same.

Line 244 ff Ecological Integrity

“As far as ecological integrity is concerned, mention should be made of indigenous 244 olive varieties such as farga, morruda, cuquello or canetera”

This is not ‘ecological integrity’  where is the ecology discussed here? Thee are ample papers on the ecology of Spanish olive orchards (such as Rey, Pedro J. "Preserving frugivorous birds in agro‐ecosystems: lessons from Spanish olive orchards." Journal of Applied Ecology 48, no. 1 (2011): 228-237. and Infante-Amate, J., 2012. The ecology and history of the Mediterranean olive grove: the Spanish great expansion, 1750-2000. Rural History, 23(2), pp.161-184.) but I suspect this not the ecology that the authors have in mind…

You are right: we have added some remarks about ecological aspects and removed some not discussing ecological issues. We have also added the mentioned references. Than you again for your remarks.

Line 255ff Visibility and visual quality

This section is methodologically unsound

We have used the terminology and description by Mayordomo-Maya, S.; Hermosilla-Pla, J. Evaluation of Landscape Quality in Valencia’s Agricultural Gardens—A Method Adapted to Multifunctional, Territorialized Agrifood Systems (MTAS). Land 2022 11(3), 398.

According to them, the criterion of visibility and visual quality considers the breadth of the visible territory, visual connectivity with other spaces and visual reach. These parameters help characterize the landscape in scenic terms. Landscape units of high visual quality are valued.

Line 196ff Social

This section is very weak . Whee is the social value here? Read Jones, S. (2017). Wrestling with the Social Value of Heritage: Problems, Dilemmas and Opportunities. Journal of Community Archaeology & Heritage, 4(1), 21-37

You are absolutely true. We changed all the paragraph, now highlighting the social role of cooperatives. Thank you again for helping us improving our article.

Line 303 Symbolic/Identity

This section is methodologically unsound

We are sorry to hear it, but for us millenary olive trees are crucial to explain the identity of this region.

Line 332          Artistic

Best to drop this. Where is the relevance?

The relevance is linked to the millenary olive trees, which are used as an artistic object because its rare originality.

Line 340          Informative/Scientific

Scientific value in cultural heritage management has very specific meanings. Not hpw the authors interpret this…

We changed all the paragraph, now highlighting the lack of scientific diffusion on the international scene about this subject. 

I am stopping the review at this point…

MINOR ISSUES

Line 8  Not everyone reading the paper will know what “Valencian Maestrat” is…so you need to explain that…as you did in line 77

Thank you for this remark. We explained it quite before in the text, as you recommend us.

The purpose of this article is to analyze the Maestrat olive growing cultural landscape, which is a district to the north of the Region of Valencia (Spain) (Figures 1–3).

Line 45/46       consider citing:

Stephenson, Janet. "The Cultural Values Model: An integrated approach to values in landscapes." Landscape and urban planning 84, no. 2 (2008): 127-139.

Cosgrove, D. (1989). Geography is everywhere: culture and symbolism in human landscapes. Horizons in human geography, 118-135

Thank you for let me know about these two references (we cited them both in the article)

Sentence ending line 61 needs references… done

Sentence ending line 81 needs a reference…done

Sentence ending line 83 needs a reference…done

Sentence ending line 92 needs a reference…done

Line 94…why “this must have been maintained” ? corrected

Sentence ending line 98 needs a reference…done

Line 100          There needs to be an academic reference that validates the dating and that cannot be the be that underpins Figure 2 done

Line 103 to 111            this assertion  needs to be referenced done

Line 113 to 121            this assertion  needs to be referenced We got this information via interviewees

Line 130          what is a ‘union farmer’ ? needs explanation Actually this expression is mistaken and the proper form is “Farmers’ Union” (a union labor of farmers) (corrected)

Figure 2           The total boundaries of the Valencia Maestrat should be shown, with the areas over 40% coverage with olive groves being shaded. You can see it better in figure 1, as figure 2 is more to show the GIAHS

Line 166 what are what are they and what is the time depth here? This this assertion also needs to be referenced. Done

LANGUAGE

The paper contains numerous infelicities in grammar and expression. It needs to be edited by a professional native-English speaking scientifically trained editor.

Line 99            what is an “empirical presence” ? Translation issue from the Spanish? The presence of olive trees is empirically proven since that time (because there are living specimens that are a thousand years old).

Line 113          ”aim of this article is to know if” done

Line 123          “This communication”  this is designated an “Article” in line 1. What is it? A communication or an article…Surely the latter thanks for noting the error

Figure 2           “olive tree occupation” thanks for noting the error (olive tree covering)

ETHICS

Given that the study relies on  semi-structured interviews with representatives of cooperatives,  union farmers and private producers, there is a need that prior ethics approval has been obtained. The authors need to provide evidence to that effect from the Ethics Review Board of their University, 

We will add this after commenting with the editor

Comments on the Quality of English Language

The paper contains numerous infelicities in grammar and expression. It needs to be edited by a professional native-English speaking scientifically trained editor.

The article was reviewed by a professional native-English speaking scientifically trained editor (We send her certificate to the Editor). 

Submission Date

18 May 2023

Date of this review

Reviewer 2 Report

1. INTRODUCTION

In line 53 et seq. reference is made to a new CAP subsidy linked to landscape conservation. It should be explained what this measure consists of.

The paragraph beginning in line 79 should be strengthened with some authoritative reference. On the other hand, from line 86 onwards I think there is a conceptual error: not all of Andalusia is so flat and fertile and not all of Valencia is so dry and stony. In this sense, the productivity of the olive grove has to do with the land it occupies and the cultivation regime, but this is something that transcends mere regional location. Thus, there are marginal olive groves in Andalusia and very productive olive groves in Valencia.

In line 92, reference is made to a possible pre-Roman origin of the olive grove in the Levantine region. It would be useful to reinforce this idea with some supporting reference.

2. METHODS AND CASE STUDY:

It is necessary to expand the information on the interviewees, both in terms of their number and, especially, their profile with respect to the olive holdings they own or the role they play in the olive oil value chain.

3. RESULTS

In the paragraph beginning in line 191 it would be convenient to provide quantitative data referring to the surface area occupied by olive groves in the study area. How many hectares are we talking about?

On the other hand, it could be deduced from the wording that the word ‘lampante’ is an anglicism in Spanish, when it has a Greek and Latin origin.

The paragraph beginning in line 229 should explain in a little more detail how the protection regulations are preventing the advance of alternative energy megaprojects.

Is what is stated in line 250 and following an opinion of the interviewees or does it have some empirical basis? I have the same doubt about what is indicated in line 291 and following. On the other hand, the CAP 2023-2027 for olive groves includes measures to compensate for environmentally and climate-friendly practices, which should be mentioned.

In line 350 and subsequent lines, I believe that the action of wind is overestimated as a factor that worsens the final quality of the oils. In any case, I suppose that this is the opinion of those interviewed, but the absence of statistical data makes me doubt it. In any case, as will be explained later, this fact should be a factor conducive to the innovation that in other places has meant bringing forward the harvest to increase quality levels, even at the expense of the quantity of oil produced. The scientific literature has made it clear that it is possible to achieve the highest quality irrespective of the variety, provided that care is taken in the field and the time between harvesting and oil extraction is reduced as much as possible.

The 365 line refers to obtaining oils of the highest quality as a basis for improving the farmer's position in the value chain. Again, quantitative information on the subject is missing.

In the section on socio-economic profitability, I believe that the information could be improved by reflecting on the role of protected designations of origin and the possibilities of concentric diversification, especially, because of the focus of the work, of olive oil tourism.

In line 406 et seq. (and this question appears at other points), the interviewed olive growers' feeling of grievance compared to that of the Andalusian growers is mentioned. To what extent has this been addressed in the reform of the CAP for the period 2023-2027?

REFERENCES

In order to address some of the questions raised, it is suggested that the following articles be consulted:

Budí, V., Rubert, J. (2020): El cultivo del olivo y la producción de aceite en la provincia de Castellón. Edita Universitat de València

Casado-Montilla, J. et al. (2023). El oleoturismo como instrumento de diversificación de las cooperativas olivícolas. REVESCO, 143, 1-17.

Rodríguez-Cohard, J. C., Sánchez Martínez, J. D. y Gallego Simón, V. J. (2017): The upgrading strategy of olive oil producers in Southern Spain: origin, development and constraints, Rural Society, 26(1): 30-47.

Rodríguez-Cohard, J. C., Sánchez-Martínez, J. D. y Gallego Simón, V. J. (2019): Olive crops and rural development: Capital, knowledge and tradition, Regional Science, Policy and Practice, 11: 935-949.

Author Response

  1. INTRODUCTION

In line 53 et seq. reference is made to a new CAP subsidy linked to landscape conservation. It should be explained what this measure consists of.

New information about CAP eco-schemes have been added in section 3.1.3 of the results (Ecological integrity). In order to prevent repetitions, we will keep the information in this section 3.1.3.

"Organic farming, although still not widely used, has recently been promoted by the new EU CAP (2023–27) eco-schemes, which offer economic rewards for maintaining a vegetation cover under the tree while it is in winter dormancy or mechanical weeding techniques that regenerate the soil and combat erosion, as opposed to the use of chemical products (Figure 4) [15]."

The paragraph beginning in line 79 should be strengthened with some authoritative reference. Done (a reference is added)

"Among Mediterranean landscapes, the olive tree, together with vines and wheat, forms a trilogy that dates back to antiquity [20]."

On the other hand, from line 86 onwards I think there is a conceptual error: not all of Andalusia is so flat and fertile and not all of Valencia is so dry and stony. In this sense, the productivity of the olive grove has to do with the land it occupies and the cultivation regime, but this is something that transcends mere regional location. Thus, there are marginal olive groves in Andalusia and very productive olive groves in Valencia.

I have modified the comment on Andalusian and Valencian soils

"On the eastern Iberian Mediterranean slopes there are some dry and stony regions, generally less productive than those of Andalusia, where the olive growing tradition dates back at least the last two millennia"

In line 92, reference is made to a possible pre-Roman origin of the olive grove in the Levantine region. It would be useful to reinforce this idea with some supporting reference.

Done (three references)

"In the agricultural system of Maestrat–a region of rough soils and dry land–olive trees have probably been cultivated since the time of the Iberians (around 500 BCE) [23–25]."

  1. METHODS AND CASE STUDY:

It is necessary to expand the information on the interviewees, both in terms of their number and, especially, their profile with respect to the olive holdings they own or the role they play in the olive oil value chain.

Thank you for your remark. Many new information on the interviewees has been added.

"Significant qualitative information was obtained from the semi-structured interviews: six people between the ages of 45 and 70 were interviewed, all of whom were linked to the Maestrat MTAS. The mean age of the interviewees was 58, which may seem very high, but it corresponds to the profile of the average farmer in the area. All of them were men: no women with an appropriate profile were found to be interviewed. The interviewees came from three different villages: La Jana, Traiguera and Canet lo Roig. In these three municipalities, olive tree crops and millenary olive trees are remarkably widespread.

Two representatives of cooperative agro-industrial systems, two private producers and two farmers who were members of a farmers’ union were interviewed. The first four interviewees were selected as representatives of the two main systems of production in Maestrat (cooperatives and private production) and the last two as members of a farmers’ union that has fought to preserve the Maestrat cultural landscape. In addition, one of the private producers interviewed is also a local development councilor in a Maestrat town council. Two of the interviewees were educated at college level, one of them had studied at an agricultural school, and the other three were educated at high school level.

The interview period was October 2022. The interview data collection method was audio recording. The interviews were conducted in-person and in their respective villages. The language used was Catalan/Valencian, which is the language commonly used by all the interviewees [31]. The questions were agreed upon by the MTAS working group in the Department of Geography at the University of Valencia. The ten questions asked were (1) Do you think that this territory conforms to a MTAS? (2) If yes, who are the main actors in this MTAS? (3) Why does this system differ from others? (4) Does this system preserve its original landscape? (5) What traditional techniques and knowledge distinguish this system from others? (6) Is the current system environmentally sustainable? Is it socioeconomically sustainable? (7) What elements beyond farming make this system a multifunctional space? (8) Do the governmental authorities make investments to preserve this system? (9) Is there social identification with this system? (10) What are the main strengths and weaknesses of this system?

The interviews were crucial to understanding the olive growing context of Maestrat. It can be seen how the interviewees describe the same issues from different perspectives, which helps to better understand their particular interests. For the cooperative leaders, cooperatives are a way of subsistence that is essential for the survival of local agriculture; for the private producers, cooperatives are a production system that prevents innovation; for the farmer’s unionists, cooperatives are a plausible system but they are currently too small and a large regional cooperative would be advisable. All of them agreed on the problems of olive growing associated with the climate and soil. Cooperative leaders were not very concerned with preserving the landscape, although they recognized that the ancient olive trees are a qualitative asset for selling their products. Private producers and trade unionists defend the landscape’s values. Cooperative members also claimed that the poor quality of the olive oil is inevitable due to climate limitations, but private producers–aware of this constraint–innovate by adapting the harvest to climate conditions and improving the quality of the oil. All the answers were very significant and helped to better understand the general situation of olive growing in the Maestrat MTAS from a qualitative viewpoint."

  1. RESULTS

In the paragraph beginning in line 191 it would be convenient to provide quantitative data referring to the surface area occupied by olive groves in the study area. How many hectares are we talking about?

Done (14.676 hectares) (rounded to 15,000)

"The total sum of hectares in our study area is close to 15,000 hectares [35]."

On the other hand, it could be deduced from the wording that the word ‘lampante’ is an anglicism in Spanish, when it has a Greek and Latin origin.

Done (new words for this paragraph)

"This olive growing landscape produces a small part of extra virgin and single-varietal oil, which is appreciated for its excellent quality; however, most of Maestrat's production is lampante oil. Lampante is a Spanish word –cf English word lamp, from Latin origin– since this kind of oil was of poor quality and mainly used for producing light in lamps."

The paragraph beginning in line 229 should explain in a little more detail how the protection regulations are preventing the advance of alternative energy megaprojects.

Done. Added text for this paragraph

"We say from the time being because this GIAHS protection does not imply that renewable energy plants will not be installed in the future, as has been seen in the very highly protected UNESCO Biosphere Reserve of Menorca (Spain), whose landscape is currently threatened by the installation of this type of projects [39]."

Is what is stated in line 250 and following an opinion of the interviewees or does it have some empirical basis?

Empirical basis (a reference has been added)

"There are contrasting differences between organic practices–based on the use of natural processes– and conventional practices, which do use chemical processes (such as fertilizers and pesticides) [41]."

I have the same doubt about what is indicated in line 291 and following.

In this case it is the opinion of interviewees (and mine).

On the other hand, the CAP 2023-2027 for olive groves includes measures to compensate for environmentally and climate-friendly practices, which should be mentioned.

They are added and mentioned in section 3.1.3

"Organic farming, although still not widely used, has recently been promoted by the new EU CAP (2023–27) eco-schemes, which offer economic rewards for maintaining a vegetation cover under the tree while it is in winter dormancy or mechanical weeding techniques that regenerate the soil and combat erosion, as opposed to the use of chemical products (Figure 4) [15]."

In line 350 and subsequent lines, I believe that the action of wind is overestimated as a factor that worsens the final quality of the oils. In any case, I suppose that this is the opinion of those interviewed, but the absence of statistical data makes me doubt it. In any case, as will be explained later, this fact should be a factor conducive to the innovation that in other places has meant bringing forward the harvest to increase quality levels, even at the expense of the quantity of oil produced. The scientific literature has made it clear that it is possible to achieve the highest quality irrespective of the variety, provided that care is taken in the field and the time between harvesting and oil extraction is reduced as much as possible.

All the interviewees coincide in the wind factor as crucial for the harvest. The cierzo or mistral winds, are, in fact, very strong in this area. Indeed, this factor has led to an earlier harvest in some cases, and although the quantity of oil has been reduced, the quality of it has increased, as you describe so well. This is explained accurately by you at your remark, and in the new text (lines 421, 459 and 525 of the reviewed text).

The 365 line refers to obtaining oils of the highest quality as a basis for improving the farmer's position in the value chain. Again, quantitative information on the subject is missing.

I do not have quantitative information about it. I do not think it is published. It is the opinion of a private producer interviewee.

In the section on socio-economic profitability, I believe that the information could be improved by reflecting on the role of protected designations of origin and the possibilities of concentric diversification, especially, because of the focus of the work, of olive oil tourism.

Done (new references added, new text added)

"It must be said there is an appellation of origin for olive oil from the Region of Valencia but the Maestrat MTAS does not use it, since it has its own label of quality linked to the thousand-year-old olive trees [50]. Around this label of quality there is economic diversification linked to olive oil and olive tree tourism [51–52], with synergies between restaurants, shops, museums, and routes visiting traditional olive groves and their dry-stone architecture [36]."

In line 406 et seq. (and this question appears at other points), the interviewed olive growers' feeling of grievance compared to that of the Andalusian growers is mentioned. To what extent has this been addressed in the reform of the CAP for the period 2023-2027?

Olive-growing Valencian farmers have no hope that this will change much with the new CAP (2023-27). As there is no information published about it, I have not added anything in the article.

REFERENCES

In order to address some of the questions raised, it is suggested that the following articles be consulted:

Budí, V., Rubert, J. (2020): El cultivo del olivo y la producción de aceite en la provincia de Castellón. Edita Universitat de València Thank you. It has been included.

Casado-Montilla, J. et al. (2023). El oleoturismo como instrumento de diversificación de las cooperativas olivícolas. REVESCO, 143, 1-17. Thank you. It has been included.

Rodríguez-Cohard, J. C., Sánchez Martínez, J. D. y Gallego Simón, V. J. (2017): The upgrading strategy of olive oil producers in Southern Spain: origin, development and constraints, Rural Society, 26(1): 30-47. Thank you. It has been included.

Rodríguez-Cohard, J. C., Sánchez-Martínez, J. D. y Gallego Simón, V. J. (2019): Olive crops and rural development: Capital, knowledge and tradition, Regional Science, Policy and Practice, 11: 935-949. Thank you. It has been included.

Thank you for all your remarks, that will improve the whole article. 

Submission Date

18 May 2023

Date of this review

29 May 2023 11:22:31

Reviewer 3 Report

I like the paper very much, but in its current form it is not suitable for publication. It is well written and easy to read, it tells a fascinating story of a landscape with all nuances, but at this moment it is not yet a scientific paper. And the major problem is that the methods described for assessing the landscape and obtaining the results are not described properly. The results look interesting, but we have no idea where did the authors get them from.

So please rewrite completely the section on methods. Give us more information both about the interviews and the assessment. How were the interviewees selected? Who they were – older men, younger women, educated in agricultural schools, … ? How they were interviewed? What did they tell? How were these different sides of the story expressed? The same with the assessment. Now you just list the used values, but tell nothing about who carried out the assessment, what were the criteria, how did you reach these results and not others. Do this rewriting in a way that another researcher elsewhere could use your steps to design his/her own project and that that researcher could find all answers to his/her possible questions in your methods section.

Also, have one more critical look at your text – there are too many repetitions. We read several times about the reasons why the outcome of oil pressing is only lampante and that the oil has characteristic spicy and bitter taste. Once is enough.

Finally, make sure your conclusions are based on your results and discussion. Now you have, for instance, ecosystem services popping up in the conclusions – nothing about them in the methods or results.

To sum it up – it is potentially a very interesting paper, but needs some work first.

No major problems with language. Text editing needed to remove repetitions

Author Response

I like the paper very much, but in its current form it is not suitable for publication. It is well written and easy to read, it tells a fascinating story of a landscape with all nuances, but at this moment it is not yet a scientific paper. And the major problem is that the methods described for assessing the landscape and obtaining the results are not described properly. The results look interesting, but we have no idea where did the authors get them from.

So please rewrite completely the section on methods. Give us more information both about the interviews and the assessment. How were the interviewees selected? Who they were – older men, younger women, educated in agricultural schools, … ? How they were interviewed? What did they tell? How were these different sides of the story expressed? The same with the assessment. Now you just list the used values, but tell nothing about who carried out the assessment, what were the criteria, how did you reach these results and not others. Do this rewriting in a way that another researcher elsewhere could use your steps to design his/her own project and that that researcher could find all answers to his/her possible questions in your methods section.

Thanks for your remarks and advice.

The methodological section referring to the interviews has been significantly expanded thanks to your plausible and helpful remarks.

"Significant qualitative information was obtained from the semi-structured interviews: six people between the ages of 45 and 70 were interviewed, all of whom were linked to the Maestrat MTAS. The mean age of the interviewees was 58, which may seem very high, but it corresponds to the profile of the average farmer in the area. All of them were men: no women with an appropriate profile were found to be interviewed. The interviewees came from three different villages: La Jana, Traiguera and Canet lo Roig. In these three municipalities, olive tree crops and millenary olive trees are remarkably widespread.

Two representatives of cooperative agro-industrial systems, two private producers and two farmers who were members of a farmers’ union were interviewed. The first four interviewees were selected as representatives of the two main systems of production in Maestrat (cooperatives and private production) and the last two as members of a farmers’ union that has fought to preserve the Maestrat cultural landscape. In addition, one of the private producers interviewed is also a local development councilor in a Maestrat town council. Two of the interviewees were educated at college level, one of them had studied at an agricultural school, and the other three were educated at high school level.

The interview period was October 2022. The interview data collection method was audio recording. The interviews were conducted in-person and in their respective villages. The language used was Catalan/Valencian, which is the language commonly used by all the interviewees [31]. The questions were agreed upon by the MTAS working group in the Department of Geography at the University of Valencia. The ten questions asked were (1) Do you think that this territory conforms to a MTAS? (2) If yes, who are the main actors in this MTAS? (3) Why does this system differ from others? (4) Does this system preserve its original landscape? (5) What traditional techniques and knowledge distinguish this system from others? (6) Is the current system environmentally sustainable? Is it socioeconomically sustainable? (7) What elements beyond farming make this system a multifunctional space? (8) Do the governmental authorities make investments to preserve this system? (9) Is there social identification with this system? (10) What are the main strengths and weaknesses of this system?

The interviews were crucial to understanding the olive growing context of Maestrat. It can be seen how the interviewees describe the same issues from different perspectives, which helps to better understand their particular interests. For the cooperative leaders, cooperatives are a way of subsistence that is essential for the survival of local agriculture; for the private producers, cooperatives are a production system that prevents innovation; for the farmer’s unionists, cooperatives are a plausible system but they are currently too small and a large regional cooperative would be advisable. All of them agreed on the problems of olive growing associated with the climate and soil. Cooperative leaders were not very concerned with preserving the landscape, although they recognized that the ancient olive trees are a qualitative asset for selling their products. Private producers and trade unionists defend the landscape’s values. Cooperative members also claimed that the poor quality of the olive oil is inevitable due to climate limitations, but private producers–aware of this constraint–innovate by adapting the harvest to climate conditions and improving the quality of the oil. All the answers were very significant and helped to better understand the general situation of olive growing in the Maestrat MTAS from a qualitative viewpoint."

As for the landscape assessment method, we have used the method by Mayordomo-Maya, S.; Hermosilla-Pla, J. Evaluation of Landscape Quality in Valencia’s Agricultural Gardens—A Method Adapted to Multifunctional, Territorialized Agrifood Systems (MTAS). Land 2022 11(3), 398. https://doi.org/10.3390/land11030398

This method is thoroughly explained in the aforementioned article. It is a method that can be discussed and improved, but the intention of this article is to use it to apply it in the Maestrat MTAS, as the authors used it to evaluate the landscape of vegetable gardens in Valencia and as other authors can apply to other territories.

Also, have one more critical look at your text – there are too many repetitions. We read several times about the reasons why the outcome of oil pressing is only lampante and that the oil has characteristic spicy and bitter taste. Once is enough.

We have simplified and refined the repetitions mentioned by you.

Finally, make sure your conclusions are based on your results and discussion. Now you have, for instance, ecosystem services popping up in the conclusions – nothing about them in the methods or results.

We have expanded the ecosystem services section in the results (subsection on ecological integrity).

"As for ecosystem services, the current almost total monoculture of olive trees in the Maestrat MTAS offers this kind of services to the non-agricultural population, for which farmers should be compensated by the political authorities for the maintenance of this non-urbanized vegetal landscape. Among these ecosystem services, it is important to mention food (olive oil), soil structure (by controlling erosion), absorption of CO2 emissions (its biomass sequestering of atmospheric CO2–which becomes part of the woody structures of plants–reduces atmospheric carbon dioxide, which accelerates climate change due to excess emissions), and hydrological services (by mitigating flood effects or recharging groundwater). The only local councillor interviewed expressed his helplessness and disappointment regarding how regional and national politicians do not support farmers for the maintenance of the landscape."

To sum it up – it is potentially a very interesting paper, but needs some work first.

Thank you very much for all your remarks, that will help to improve the whole of this article.

Comments on the Quality of English Language

No major problems with language. Text editing needed to remove repetitions

Submission Date

18 May 2023

Date of this review

05 Jun 2023 11:01:18

Round 2

Reviewer 1 Report

Second review of: The precarious survival of an ancient cultural landscape: the thousand-year-old olive trees of the Valencian Maestrat (Spain)

The authors are to be strongly commended to have made a major effort in addressing the comments of the first review round. The paper is very much improved, but still needs some further work before it can be accepted.

-------------------------------

IN MY FIRST REVIEW I WROTE:

Line 145 ff       The methodology uses the term of ‘intrinsic’ values but its discussion makes it very clear that all values are in fact not independent of context but are cultural constructs and projections of community and decision makers …”

TO THIS THE AUTHORS RESPONDED:

We have used the terminology used by Mayordomo-Maya, S.; Hermosilla-Pla, J. Evaluation of Landscape Quality in Valencia’s Agricultural Gardens—A Method Adapted to Multifunctional, Territorialized Agrifood Systems (MTAS). Land 2022 11(3), 398. https://doi.org/10.3390/land11030398

Intrinsic means “belonging to the essential nature or constitution of a thing”, and therefore it refers, by extension, to the particular characteristics of our study area.

TO THAT I HAVE THE FOLLOWING, ADDITIONAL COMMENTS:

No it does not. Characteristics yes, but VALUES, not. These are culturally conditioned human constructs that are projected in innate objects and places.

-------------------------------

IN MY FIRST REVIEW I WROTE:

Line 255ff Visibility and visual quality

This section is methodologically unsound

TO THIS THE AUTHORS RESPONDED:

We have used the terminology and description by Mayordomo-Maya, S.; Hermosilla-Pla, J. Evaluation of Landscape Quality in Valencia’s Agricultural Gardens—A Method Adapted to Multifunctional, Territorialized Agrifood Systems (MTAS). Land 2022 11(3), 398.

According to them, the criterion of visibility and visual quality considers the breadth of the visible territory, visual connectivity with other spaces and visual reach. These parameters help characterize the landscape in scenic terms. Landscape units of high visual quality are valued.

TO THAT I HAVE THE FOLLOWING, ADDITIONAL COMMENTS:

If you want to use their definition and approach, then please spell out the criteria. Adapt the sentence used in your response and add it to the text 

-------------------------------

IN MY FIRST REVIEW I WROTE:

Line 303 Symbolic/Identity

This section is methodologically unsound

TO THIS THE AUTHORS RESPONDED:

We are sorry to hear it, but for us millenary olive trees are crucial to explain the identity of this region.

TO THAT I HAVE THE FOLLOWING, ADDITIONAL COMMENTS:

There needs to an anchoring of the text in the theoretical literature on symbolic values and identity before commenting on their manifestation in the local case study. That is lacjing and that is why the section is still methodologically unsound.

-------------------------------

IN MY FIRST REVIEW I WROTE:

Line 332          Artistic

Best to drop this. Where is the relevance?

TO THIS THE AUTHORS RESPONDED:

The relevance is linked to the millenary olive trees, which are used as an artistic object because its rare originality.

TO THAT I HAVE THE FOLLOWING, ADDITIONAL COMMENTS:

In standard heritage assessments that would be a subset of social value. If you want to keep artistic value as its own section, you will to, again, anchoring the text in the theoretical literature on artistic values.

-------------------------------

IN MY FIRST REVIEW I WROTE:

Given that the study relies on  semi-structured interviews with representatives of cooperatives,  union farmers and private producers, there is a need that prior ethics approval has been obtained. The authors need to provide evidence to that effect from the Ethics Review Board of their University.

TO THIS THE AUTHORS RESPONDED:

We will add this after commenting with the editor.

TO THAT I HAVE THE FOLLOWING, ADDITIONAL COMMENTS:

Institutional Review Board Statement: Not applicable. 

Informed Consent Statement: Not applicable.

You either have formal ethics approval or you do not. If you do, add this in the relevant section in the back matter to the paper, as required by the journal. If you do not, modern ethics standards require that the paper MUST not be published. 

At this point you even claim that this is not required !! Your comments in the back matter section state:

Author Response

Second review of: The precarious survival of an ancient cultural landscape: the thousand-year-old olive trees of the Valencian Maestrat (Spain)

The authors are to be strongly commended to have made a major effort in addressing the comments of the first review round. The paper is very much improved, but still needs some further work before it can be accepted.

-------------------------------

IN MY FIRST REVIEW I WROTE:

Line 145 ff       The methodology uses the term of ‘intrinsic’ values but its discussion makes it very clear that all values are in fact not independent of context but are cultural constructs and projections of community and decision makers …”

TO THIS THE AUTHORS RESPONDED:

We have used the terminology used by Mayordomo-Maya, S.; Hermosilla-Pla, J. Evaluation of Landscape Quality in Valencia’s Agricultural Gardens—A Method Adapted to Multifunctional, Territorialized Agrifood Systems (MTAS). Land 2022 11(3), 398. https://doi.org/10.3390/land11030398

Intrinsic means “belonging to the essential nature or constitution of a thing”, and therefore it refers, by extension, to the particular characteristics of our study area.

TO THAT I HAVE THE FOLLOWING, ADDITIONAL COMMENTS:

No it does not. Characteristics yes, but VALUES, not. These are culturally conditioned human constructs that are projected in innate objects and places.

WITH YOUR PERMISSION, WE HAVE INCORPORATED YOUR ACCURATE AND PRECISE REMARKS –WITH OTHER WORDS- INTO THE TEXT.

ADDED PARAGRAPH:

Intrinsic values are cultural constructs that are projected onto inanimate objects and places; they refer to the inherent characteristics of the landscape itself, which are subjective and can mutate over time

-------------------------------

IN MY FIRST REVIEW I WROTE:

Line 255ff Visibility and visual quality

This section is methodologically unsound

TO THIS THE AUTHORS RESPONDED:

We have used the terminology and description by Mayordomo-Maya, S.; Hermosilla-Pla, J. Evaluation of Landscape Quality in Valencia’s Agricultural Gardens—A Method Adapted to Multifunctional, Territorialized Agrifood Systems (MTAS). Land 2022 11(3), 398.

According to them, the criterion of visibility and visual quality considers the breadth of the visible territory, visual connectivity with other spaces and visual reach. These parameters help characterize the landscape in scenic terms. Landscape units of high visual quality are valued.

TO THAT I HAVE THE FOLLOWING, ADDITIONAL COMMENTS:

If you want to use their definition and approach, then please spell out the criteria. Adapt the sentence used in your response and add it to the text 

THANKS FOR YOUR REMARK. WE HAVE ADAPTED THE CRITERIA ON VISIBILITY AND VISUAL QUALITY DESCRIBED BY THE MENTIONED AUTHORS

ADDED PARAGRAPH:

This refer to the breadth of the observable territory, visual connectivity with other territories and visual range. These criteria help to describe the landscape from a scenic perspective [30]. 

-------------------------------

IN MY FIRST REVIEW I WROTE:

Line 303 Symbolic/Identity

This section is methodologically unsound

TO THIS THE AUTHORS RESPONDED:

We are sorry to hear it, but for us millenary olive trees are crucial to explain the identity of this region.

TO THAT I HAVE THE FOLLOWING, ADDITIONAL COMMENTS:

There needs to an anchoring of the text in the theoretical literature on symbolic values and identity before commenting on their manifestation in the local case study. That is lacjing and that is why the section is still methodologically unsound.

THANKS FOR YOUR REMARK. WE HAVE ADAPTED THE CRITERIA ON SYMBOLIC/IDENTITY VALUES DESCRIBED BY THE MENTIONED AUTHORS

ADDED PARAGRAPH:

This refers to the emotional ties and perceptions from local people as far as landscape is concerned. It considers symbolic bonds and identity perceptions [30].

-------------------------------

IN MY FIRST REVIEW I WROTE:

Line 332          Artistic

Best to drop this. Where is the relevance?

TO THIS THE AUTHORS RESPONDED:

The relevance is linked to the millenary olive trees, which are used as an artistic object because its rare originality.

TO THAT I HAVE THE FOLLOWING, ADDITIONAL COMMENTS:

In standard heritage assessments that would be a subset of social value. If you want to keep artistic value as its own section, you will to, again, anchoring the text in the theoretical literature on artistic values.

THANKS FOR YOUR REMARK. WE HAVE ADAPTED THE CRITERIA ON ARTISTIC VALUES DESCRIBED BY THE MENTIONED AUTHORS

ADDED PARAGRAPH:

This considers artistic expressions linked to the landscape. Landscape can be represented through different artistic languages [30].

-------------------------------

IN MY FIRST REVIEW I WROTE:

Given that the study relies on  semi-structured interviews with representatives of cooperatives,  union farmers and private producers, there is a need that prior ethics approval has been obtained. The authors need to provide evidence to that effect from the Ethics Review Board of their University.

TO THIS THE AUTHORS RESPONDED:

We will add this after commenting with the editor.

TO THAT I HAVE THE FOLLOWING, ADDITIONAL COMMENTS:

Institutional Review Board Statement: Not applicable. 

Informed Consent Statement: Not applicable.

You either have formal ethics approval or you do not. If you do, add this in the relevant section in the back matter to the paper, as required by the journal. If you do not, modern ethics standards require that the paper MUST not be published. 

At this point you even claim that this is not required !! Your comments in the back matter section state:

ACCORDING TO THE PUBLISHED INFORMATION ON THE WEBPAGE OF UNIVERSITY OF VALENCIA:

https://www.uv.es/ethical-commission-experimental-research/en/ethics-research-humans/frequently-asked-questions.html

What kind of works require the approval of the Committee of Ethics and Human Research?

Generally, authorisation is required for works in which information related to health, clinical data, supplementary examinations, analytical and radiological data, treatments, etc. is involved. Having the informed consent of the diagnostic tests and the therapeutic procedures received is not sufficient. It is also necessary to include an informed consent signed by the patient or their legal representative in order to use the data, whether it is an observational or experimental research, and in any case it must be positively evaluated by the Committee of Ethics and Human Research.

THIS IS NOT THE CASE FOR OUR STUDY, WHICH IS NOT MEDICAL.

IN THE WORK BY Martínez-Arnáiz, M.; Baraja-Rodríguez, E.; Herrero-Luque, D. Multifunctional Territorialized Agri-Food Systems, Geographical Quality Marks and Agricultural Landscapes: The Case of Vineyards. Land 2022, 11(4), 457. https://doi.org/10.3390/land11040457

THEY SAID THAT “The fieldwork consisted of open-format, non-systematized interviews with members of the Regulatory Councils and winemakers of the PDO of Castilla y León (Ribera de Duero, Rueda, Toro, Bierzo), who provided valuable qualitative information for the analysis.”, BUT IN THEIR ARTICLE’S SECTIONS INSTITUTIONAL REVIEW BOARD STATEMENT AND INFORMED CONSENT STATEMENT IT SAYS NOT APPLICABLE.

IF YOU PROPOSE US A FORMULA FOR THIS SECTION WE WILL ADD IT, BUT ALL THE INSTITUTIONAL REVIEW BOARD STATEMENT THAT WE HAVE FOUND IS ABOUT MEDICAL ISSUES

THANK YOU FOR YOUR HELP AND WE APOLOGIZE FOR OUR MISTAKES

Reviewer 3 Report

The paper has been significantly improved and most of the previously missing information is now present. That means it is time to give it a structure. I am still talking about the methods section, and consequently, the results section as well. What I suggest:

1.      1. Generally, keep separately the description of the interview contents and the results what the respondents told you.

2.      2. Keep the first paragraph of the Methods section in place, follow that with the one that describes what questions were asked (appr lines 158-174), how the respondents were selected and how the interviews were carried out. The rest – what did they tell you and why – is not a description of the method, but rather a result of your interview study. Move these parts to the Results (or even discussion) sections.

3.      3. As for landscape values assessment, you still do not describe who assesses the values. Table 1 and lines 216-226 are very good and informative – now please follow these with a clarification of how you got the results as presented using your methods. Was it so that you asked the interviewees to assess these values? Did you do this assessment by yourself using the materials you got from interviews, literature studies and fieldwork? Any other option? What I want is 1-2 sentences explaining what you in fact did. I also checked the reference for the method, and that one too does not explain who is supposed to carry out the assessment, just lists the criteria. This must be clarified.

4.      4. Start the Results section with describing the outcome of your interviews (basically lines 175-197 and 141-157 that are not description of the METHOD). And then make sure the reader understands clearly where are the results derived from. The missing link between the method and the results is the weakest point of this paper – which is still a very good read.

No comments :)

Author Response

Comments and Suggestions for Authors

The paper has been significantly improved and most of the previously missing information is now present. That means it is time to give it a structure. I am still talking about the methods section, and consequently, the results section as well. What I suggest:

  1. 1. Generally, keep separately the description of the interview contents and the results what the respondents told you.

WE HAVE DONE IT AS EXPLAINED BENEATH, FOLLOWING YOUR INSTRUCTIONS

  1. Keep the first paragraph of the Methods section in place, follow that with the one that describes what questions were asked (appr lines 158-174), how the respondents were selected and how the interviews were carried out. The rest – what did they tell you and why – is not a description of the method, but rather a result of your interview study. Move these parts to the Results (or even discussion) sections.

THANK YOU VERY MUCH FOR YOUR HELP. YOU ARE ABSOLUTELY RIGHT: IT IS NOT CORRECT TO MENTION THE OUTCOME OF THE INTERVIEWS IN THE METHODS. THIS IS THE NEW STRUCTURE OF THE INTERVIEWS’ METHOD, FOLLOWING YOUR INSTRUCTIONS:

“Significant qualitative information was obtained from the semi-structured inter-views. The questions were agreed upon by the MTAS working group in the Department of Geography at the University of Valencia. The ten questions asked were (1) Do you think that this territory conforms to a MTAS? (2) If yes, who are the main actors in this MTAS? (3) Why does this system differ from others? (4) Does this system preserve its original landscape? (5) What traditional techniques and knowledge distinguish this system from others? (6) Is the current system environmentally sustainable? Is it socio-economically sustainable? (7) What elements beyond farming make this system a multifunctional space? (8) Do the governmental authorities make investments to preserve this system? (9) Is there social identification with this system? (10) What are the main strengths and weaknesses of this system?

Six people between the ages of 45 and 70 were interviewed, all of whom were linked to the Maestrat MTAS. The mean age of the interviewees was 58, which may seem very high, but it corresponds to the profile of the average farmer in the area. All of them were men: no women with an appropriate profile were found to be inter-viewed. The interviewees came from three different villages: La Jana, Traiguera and Canet lo Roig. In these three municipalities, olive tree crops and millenary olive trees are remarkably widespread.

Two representatives of cooperative agro-industrial systems, two private producers and two farmers who were members of a farmers’ union were interviewed. The first four interviewees were selected as representatives of the two main systems of production in Maestrat (cooperatives and private production) and the last two as members of a farmers’ union that has fought to preserve the Maestrat cultural landscape. In addition, one of the private producers interviewed is also a local development councilor in a Maestrat town council. Two of the interviewees were educated at college level, one of them had studied at an agricultural school, and the other three were educated at high school level.

The interview period was October 2022. The interview data collection method was audio recording. The interviews were conducted in-person and in their respective villages. The language used was Catalan/Valencian, which is the language commonly used by all the interviewees [31].”

THE INTERVIEWEES’ OPINIONS HAVE BEEN INCLUDED IN THE STARTING SECTION OF THE RESULTS, AS RECOMMENDED BY YOU

  1. 3. As for landscape values assessment, you still do not describe who assesses the values. Table 1 and lines 216-226 are very good and informative – now please follow these with a clarification of how you got the results as presented using your methods. Was it so that you asked the interviewees to assess these values? Did you do this assessment by yourself using the materials you got from interviews, literature studies and fieldwork? Any other option? What I want is 1-2 sentences explaining what you in fact did. I also checked the reference for the method, and that one too does not explain who is supposed to carry out the assessment, just lists the criteria. This must be clarified.

AS YOU VERY PROPERLY SAY, THE METHOD FOR THE ASSESSMENT OF THE LANDSCAPE AND HERITAGE QUALITY DERIVES FROM INTERVIEWS, LITERATURE REVIEW AND FIELDWORK. THIS INFORMATION HAS BEEN ADDED IN THE TEXT:

  1. Results

The method for analyzing landscape and heritage quality described in the methodology [30] assesses the intrinsic values, the heritage values, and the potential and viability values of a given territory. This assessment was made using the materials from interviews, literature studies and fieldwork.

  1. 4. Start the Results section with describing the outcome of your interviews (basically lines 175-197 and 141-157 that are not description of the METHOD). And then make sure the reader understands clearly where are the results derived from. The missing link between the method and the results is the weakest point of this paper – which is still a very good read.

WE HAVE ADDED THE OUTCOME OF OUR INTERVIEWS IN THE STARTING SECTION OF THE RESULTS.

Interviews were a crucial qualitative tool to assessing the results on the Maestrat olive growing MTAS. It can be seen how the interviewees describe the same issues from different perspectives, which helps to better understand their particular interests. For the cooperative leaders, cooperatives are a way of subsistence that is essential for the survival of local agriculture; for the private producers, cooperatives are a production system that prevents innovation; for the farmer’s unionists, cooperatives are a plausible system but they are currently too small and a large regional cooperative would be advisable. All of them agreed on the problems of olive growing associated with the climate and soil. Cooperative leaders were not very concerned with preserving the landscape, although they recognized that the ancient olive trees are a qualitative asset for selling their products. Private producers and trade unionists defend the landscape’s values. Cooperative members also claimed that the poor quality of the olive oil is inevitable due to climate limitations, but private producers–aware of this constraint–innovate by adapting the harvest to climate conditions and improving the quality of the oil. 

THANK YOU FOR YOUR HELP AND WE APOLOGIZE FOR OUR MISTAKES

Comments on the Quality of English Language

No comments :)

Round 3

Reviewer 1 Report

The authors, have addressed my concerns to such a level that the paper can be published.

Reviewer 3 Report

I think the methods part is finally mostly OK. The rest was OK already earlier. 

Some very minor issues with the last amendments, that could be corrected in proof-reading (typical: this refer to ... should be this refers to)